# Effective radiative forcing in the aerosol–climate model CAM5.3-MARC-ARG

Benjamin S. Grandey[1], Daniel Rothenberg[2], Alexander Avramov[2,3], Qinjian Jin[2], Hsiang-He Lee[1], Xiaohong Liu[4], Zheng Lu[4], Samuel Albani[5,6], and Chien Wang[2,1]

[1]Center for Environmental Sensing and Modeling, Singapore-MIT Alliance for Research and Technology, Singapore
[2]Center for Global Change Science, Massachusetts Institute of Technology, Cambridge, Massachusetts, USA
[3]Department of Environmental Sciences, Emory University, Atlanta, Georgia, USA
[4]Department of Atmospheric Science, University of Wyoming, Laramie, Wyoming, USA
[5]Department of Earth and Atmospheric Sciences, Cornell University, Ithaca, New York, USA
[6]Laboratoire des Sciences du Climat et de l'Environnement, LSCE/IPSL, CEA-CNRS-UVSQ, Gif-sur-Yvette, France

*Correspondence to*: Benjamin S. Grandey (benjamin@smart.mit.edu)

**Abstract.** We quantify the effective radiative forcing (ERF) of anthropogenic aerosols modelled by the aerosol–climate model CAM5.3-MARC-ARG. CAM5.3-MARC-ARG is a new configuration of the Community Atmosphere Model version 5.3 (CAM5.3) in which the default aerosol module has been replaced by the two-Moment, Multi-Modal, Mixing-state-resolving Aerosol model for Research of Climate (MARC). CAM5.3-MARC-ARG uses the ARG aerosol activation scheme, consistent with the default configuration of CAM5.3. We compute differences between simulations using year-1850 aerosol emissions and simulations using year-2000 aerosol emissions in order to assess the radiative effects of anthropogenic aerosols. We compare the aerosol lifetimes, aerosol column burdens, cloud properties, and radiative effects produced by CAM5.3-MARC-ARG with those produced by the default configuration of CAM5.3, which uses the modal aerosol module with three lognormal modes (MAM3), and a configuration using the modal aerosol module with seven lognormal modes (MAM7). Compared with MAM3 and MAM7, we find that MARC produces stronger cooling via the direct radiative effect, the shortwave cloud radiative effect, and the surface albedo radiative effect; similarly, MARC produces stronger warming via the longwave cloud radiative effect. Overall, MARC produces a global mean net ERF of $-1.79 \pm 0.03$ W m$^{-2}$, which is stronger than the global mean net ERF of $-1.57 \pm 0.04$ W m$^{-2}$ produced by MAM3 and $-1.53 \pm 0.04$ W m$^{-2}$ produced by MAM7. The regional distribution of ERF also differs between MARC and MAM3, largely due to differences in the regional distribution of the shortwave cloud radiative effect. We conclude that the specific representation of aerosols in global climate models, including aerosol mixing state, has important implications for climate modelling.

## 1 Introduction

Aerosol particles influence the earth's climate system by perturbing its radiation budget. There are three primary mechanisms by which aerosols interact with radiation. First, aerosols interact directly with radiation by scattering and

absorbing solar and thermal infrared radiation (Haywood and Boucher, 2000). Second, aerosols interact indirectly with radiation by perturbing clouds, by acting as the cloud condensation nuclei on which cloud droplets form and the ice nuclei that facilitate freezing of cloud droplets (Fan et al., 2016; Rosenfeld et al., 2014): for example, an aerosol-induced increase in cloud cover would lead to increased scattering of "shortwave" solar radiation and increased absorption of "longwave"

thermal infrared radiation. Third, aerosols can influence the albedo of the earth's surface: for example, deposition of absorbing aerosol on snow reduces the albedo of the snow, causing more solar radiation to be absorbed at the earth's surface (Jiao et al., 2014).

The "effective radiative forcing" (ERF) of anthropogenic aerosols, defined as the top-of-atmosphere radiative effect caused by anthropogenic emissions of aerosols and aerosol precursors, is often used to quantify the radiative effects of

aerosols (Boucher et al., 2013). In contrast to instantaneous radiative forcing, ERF allows rapid adjustments – including changes to clouds – to occur (Sherwood et al., 2015). The anthropogenic aerosol ERF is approximately equivalent to "the radiative flux perturbation associated with a change from preindustrial to present-day [aerosol emissions], calculated in a global climate model using fixed sea surface temperature" (Haywood et al., 2009). This approach "allows clouds to respond to the aerosol while [sea] surface temperature is prescribed" (Ghan, 2013).

The primary tools available for investigating the anthropogenic aerosol ERF are state-of-the-art global climate models. However, there is widespread disagreement among these models, especially regarding the magnitude of anthropogenic aerosol ERF (Quaas et al., 2009; Shindell et al., 2013). The magnitude of the ERF of anthropogenic aerosols is highly uncertain: estimates of the global mean anthropogenic aerosol ERF range from $-1.9$ to $-0.1$ W m$^{-2}$ (Boucher et al., 2013). Much of this uncertainty can be attributed to uncertainty in pre-industrial aerosol emissions (Carslaw et al., 2013).

Model parameterizations constitute another large source of uncertainty. Of particular importance are model parameterizations relating to aerosol–cloud interactions, such as the aerosol activation scheme (Rothenberg et al., 2018), the choice of autoconversion threshold radius (Golaz et al., 2011), and constraints on the minimum cloud droplet number concentration (Hoose et al., 2009). The detailed representation of aerosols also likely plays an important role, because the aerosol particle size and chemical composition determine hygroscopicity and hence influence aerosol activation (Petters and

Kreidenweis, 2007).

The present-day anthropogenic aerosol ERF partially masks the warming effects of anthropogenic greenhouse gases. Therefore, the large uncertainty in the anthropogenic aerosol ERF is a major source of uncertainty in estimates of equilibrium climate sensitivity and projections of future climate (Andreae et al., 2005). Furthermore, the anthropogenic aerosol ERF is regionally inhomogeneous, adding another source of uncertainty in climate projections (Shindell, 2014). The

regional inhomogeneity of the anthropogenic aerosol ERF has likely also influenced rainfall patterns during the 20th century (Wang, 2015). In order to improve understanding of current and future climate, including rainfall patterns, it is necessary to improve understanding of the magnitude and regional distribution of the anthropogenic aerosol ERF.

In this manuscript, we investigate the uncertainty in anthropogenic aerosol ERF associated with the representation of aerosols in global climate models. In particular, we assess the aerosol radiative effects produced by a new configuration

of the Community Atmosphere Model version 5.3 (CAM5.3). In this new configuration – CAM5.3-MARC-ARG – the default modal aerosol module has been replaced with the two-Moment, Multi-Modal, Mixing-state-resolving Aerosol model for Research of Climate (MARC). We compare the aerosol fields and aerosol radiative effects produced by CAM5.3-MARC-ARG with those produced by the default modal aerosol module in CAM5.3.

## 2 Methodology

### 2.1 Modal aerosol modules (MAM3 and MAM7)

The Community Earth System Model version 1.2.2 (CESM 1.2.2) contains the Community Atmosphere Model version 5.3 (CAM5.3). Within CAM5.3, the default aerosol module is a modal aerosol module which parameterizes the aerosol size distribution using three lognormal modes (MAM3), each assuming a total internal mixture of a set of fixed chemical species (Liu et al., 2012). Optionally, a more detailed modal aerosol module with seven lognormal modes (MAM7) (Liu et al., 2012) can be used instead of MAM3. More recently, a version containing four modes (MAM4) (Liu et al., 2016) has also been coupled to CAM5.3, but we do not consider MAM4 in this study.

The seven modes included in MAM7 are Aitken, accumulation, primary carbon, fine soil dust, fine sea salt, coarse soil dust, and coarse sea salt. Depending on the mode, MAM7 simulates the mass mixing ratios of internally-mixed sulfate, ammonium, primary organic matter, secondary organic matter, black carbon, soil dust, and sea salt (Liu et al., 2012).

In MAM3, four simplifications are made: first, the primary carbon mode is merged into the accumulation mode; second, the fine soil dust and fine sea salt modes are also merged into the accumulation mode; third, the coarse soil dust and coarse sea salt modes are merged to form a single coarse mode; and fourth, ammonium is implicitly included via sulfate and is no longer explicitly simulated. As a result, MAM3 simulates just three modes: Aitken, accumulation, and coarse. This reduces the computational expense of the model.

In this manuscript, we often refer to MAM3 and MAM7 collectively as "MAM". The MAM-simulated aerosols interact with radiation, allowing aerosol direct and semi-direct effects to be represented. The aerosols can act as cloud condensation nuclei via the ARG aerosol activation scheme (Abdul-Razzak and Ghan, 2000); sulfate and dust also act as ice nuclei. Via such activation, the aerosols are coupled to the stratiform cloud microphysics (Gettelman et al., 2010; Morrison and Gettelman, 2008), allowing aerosol indirect effects on stratiform clouds to be represented. These indirect effects dominate the anthropogenic aerosol ERF in CAM version 5.1 (CAM5.1) (Ghan et al., 2012). In comparison with many other global climate models, the anthropogenic aerosol ERF in CAM5.1 is relatively strong (Shindell et al., 2013).

### 2.2 The two-Moment, Multi-Modal, Mixing-state-resolving Aerosol model for Research of Climate (MARC)

The two-Moment, Multi-Modal, Mixing-state-resolving Aerosol model for Research of Climate (MARC), which is based on the aerosol microphysical scheme developed by Ekman et al. (2004, 2006) and Kim et al. (2008), simulates the evolution of mixtures of aerosol species. Previous versions of MARC have been used both in cloud-resolving model simulations (Ekman

et al., 2004, 2006, 2007; Engström et al., 2008; Wang, 2005a, 2005b) and in global climate model simulations (Kim et al., 2008, 2014; Ekman et al., 2012). Recently, an updated version of MARC has been coupled to CAM5.3 within CESM1.2.2 (Rothenberg et al., 2018).

In contrast to MAM, MARC tracks the number concentrations and mass concentrations of both externally-mixed and internally-mixed aerosol modes with assumed lognormal size distributions. The externally-mixed modes include three pure sulfate modes (nucleation, Aitken, and accumulation), pure organic carbon (OC), and pure black carbon (BC). The internally-mixed modes include mixed organic carbon plus sulfate (MOS) and mixed black carbon plus sulfate (MBS). In the MOS mode, it is assumed that the organic carbon and sulfate are mixed homogeneously within each particle; in the MBS mode, it is assumed that each particle contains a black carbon core surrounded by a sulfate shell. Sea salt and mineral dust are represented using sectional single-moment schemes, each with four size bins (Albani et al., 2014; Mahowald et al., 2006; Scanza et al., 2015). Table 1 of Rothenberg et al. (2018) contains details of the size distribution, density, and hygroscopicity of each mode. It is assumed that the pure OC and pure BC modes are hydrophobic (Petters and Kreidenweis, 2007; Rothenberg et al., 2018).

Sea salt emissions follow the default scheme used by MAM (Liu et al., 2012), based on simulated wind speed and sea surface temperature. Dust emissions follow the tuning of Albani et al. (Albani et al., 2014), based on simulated wind speed and soil properties, including soil moisture and vegetation cover. Emissions of sulfur dioxide, dimethyl sulfide, sulfate (as gas-phase sulfuric acid), organic carbon aerosol, black carbon aerosol, and volatile organic compounds (such as isoprene and monoterpene) are prescribed. The volatile organic compounds are converted upon emission into pure OC.

The aerosol removal processes represented by MARC – including nucleation scavenging by both stratiform and convective clouds, impaction scavenging by precipitation, and dry deposition – are based on aerosol size and mixing state. Evaporation of cloud and rain drops results in resuspension of sulfate aerosol in the accumulation-mode.

Figure 1 summarises the physical and chemical processes represented by MARC. Further details about the formulation of MARC, as well as validation of its simulated aerosol fields compared with observations, can be found in the body and Supplement of Rothenberg et al. (2018).

Whereas previous versions of MARC represented only direct interactions between aerosols and radiation (Kim et al., 2008), an important feature of the new version of MARC is that the aerosols also interact indirectly with radiation via clouds. The MARC-simulated aerosols interact with stratiform cloud microphysics via the default stratiform cloud microphysics scheme (Gettelman et al., 2010; Morrison and Gettelman, 2008), as would be the case for the default MAM3 configuration of CAM5.3. Various aerosol activation schemes can be used with MARC (Rothenberg et al., 2018), including versions of a recently-developed scheme based on polynomial chaos expansion (Rothenberg and Wang, 2016, 2017). The choice of activation scheme can substantially influence the ERF (Rothenberg et al., 2018) In order to facilitate comparison between the MAM and MARC aerosol modules, we have chosen to keep the activation scheme constant in this study: as is the case for the MAM simulations, the ARG activation scheme (Abdul-Razzak and Ghan, 2000) is also used for the MARC simulations. We refer to this configuration as "CAM5.3-MARC-ARG".

**2.3 Simulations**

In order to compare results from MAM3, MAM7, and MARC, six CAM5.3 simulations are performed:

1. "MAM3_2000", which uses MAM3 with year-2000 aerosol (including aerosol precursor) emissions;
2. "MAM7_2000", which uses MAM7 with year-2000 aerosol emissions;
3. "MARC_2000", which uses MARC with year-2000 aerosol emissions;
4. "MAM3_1850", which uses MAM3 with year-1850 aerosol emissions;
5. "MAM7_1850", which uses MAM7 with year-1850 aerosol emissions; and
6. "MARC_1850", which uses MARC with year-1850 aerosol emissions.

The three simulations using year-2000 emissions, referred to as the "year-2000 simulations", facilitate comparison of aerosol
fields and cloud fields; the three simulations using year-1850 emissions, referred to as the "year-1850 simulations", further facilitate analysis of the aerosol radiative effects produced by MAM and MARC. In the figures and discussion of results, "2000-1850" and "Δ" both refer to differences between the year-2000 simulation and the year-1850 simulation for a given aerosol module (e.g. MARC_2000-MARC_1850). The "2000-1850" differences should be interpreted as aerosol-induced differences, arising due to changes in aerosol emissions alone: the only difference between the year-2000 simulations and the
year-1850 simulations is the aerosol (including aerosol precursor) emissions.

In this study, we deliberately use identical emissions for MAM and MARC so that the influence of emissions inventories can be minimised when the results are compared. The prescribed emissions for both MAM and MARC follow the default MAM emissions files, described in the Supplement of Liu et al. (2012), based on Lamarque et al. (2010). (The ammonia emissions for the year-1850 MAM7 simulation are an exception, being based on the year-1850 ammonia emissions
for the Coupled Model Intercomparison Project Phase 6.) This differs from Rothenberg et al. (2018), who used different emissions of organic carbon aerosol, black carbon aerosol, and volatile organic compounds.

For the MAM simulations, the emissions from forest fires and grass fires follow a vertical profile, as do sulfur emissions from the energy and industry sectors (Liu et al., 2012). For the MARC simulations, sulfur emissions follow the same vertical profile as for MAM; but all organic carbon, black carbon, and volatile organic compounds are emitted at the
surface. 2.5% of the sulfur dioxide is emitted as primary sulfate. The yield coefficients for conversion of volatile organic compounds to organic carbon follow Liu et al. (2012). For the MAM simulations, the organic carbon aerosol emissions are converted to primary organic matter emissions using a scale factor of 1.4; for the MARC simulations, no such scale factor is applied. Mineral dust and sea salt emissions are not prescribed, being calculated "online".

CESM 1.2.2, with CAM5.3, is used for all simulations. Greenhouse gas concentrations and sea surface
temperatures (SSTs) are prescribed using year-2000 climatological values, based on the "F_2000_CAM5" component set. CAM5.3 is run at a horizontal resolution of 1.9°×2.5° with 30 levels in the vertical direction. Clean-sky radiation diagnostics are included, facilitating diagnosis of the direct radiative effect. The Cloud Feedback Model Intercomparison

Project (CFMIP) Observational Simulator Package (COSP) (Bodas-Salcedo et al., 2011) is switched on, although the COSP diagnostics are not analysed in this manuscript.

Each simulation is run for 32 years, and the first two years are excluded as spin-up. Hence, a period of 30 years is analysed. The ensemble of 30 annual means can be used to assess significance and calculate standard errors.

## 2.4 Diagnosis of radiative effects

Pairs of prescribed-SST simulations, with differing aerosol emissions, facilitate diagnosis of anthropogenic aerosol ERF via the "radiative flux perturbation" approach (Haywood et al., 2009; Lohmann and Feichter, 2005). When "clean-sky" radiation diagnostics are available, the ERF can be decomposed into contributions from different radiative effects (Ghan, 2013). (We use the term "radiative forcing" only when referring to ERF, defined as the radiative flux perturbation between a simulation using year-1850 emissions and a simulation using year-2000 emissions; we use the term "radiative effect" more generally.)

Following Ghan (2013), the shortwave effective radiative forcing ($ERF_{SW}$) can be decomposed as follows:

$$ERF_{SW} = \Delta DRE_{SW} + \Delta CRE_{SW} + \Delta SRE_{SW} \tag{1}$$

where $\Delta$ refers to the 2000-1850 difference, $DRE_{SW}$ is the direct radiative effect, $CRE_{SW}$ is the clean-sky shortwave cloud radiative effect, and $\Delta SRE_{SW}$ is the 2000-1850 surface albedo radiative effect. These components are defined as follows:

$$ERF_{SW} = \Delta F \tag{2}$$
$$DRE_{SW} = (F - F_{clean}) \tag{3}$$
$$CRE_{SW} = (F_{clean} - F_{clean,clear}) \tag{4}$$
$$\Delta SRE_{SW} = \Delta F_{clean,clear} \tag{5}$$

where $F$ is the net shortwave flux at top-of-atmosphere (TOA), $F_{clean}$ is the clean-sky net shortwave flux at TOA, and $F_{clean,clear}$ is the clean-sky clear-sky net shortwave flux at TOA. ("Clear-sky" refers to a hypothetical situation where clouds do not interact with radiation; "clean-sky" refers to a hypothetical situation where aerosols do not directly interact with radiation.)

The longwave effective radiative forcing ($ERF_{LW}$) is calculated as follows:

$$ERF_{LW} = \Delta L \approx \Delta(L - L_{clear}) = \Delta CRE_{LW} \tag{6}$$

where $L$ is the net longwave flux at TOA, $L_{clear}$ is the clear-sky net longwave flux at TOA, and $CRE_{LW}$ is the longwave cloud radiative effect. Eq. (6) assumes that aerosols and surface albedo changes do not influence the longwave flux at TOA, so that $\Delta L_{clear} \approx 0$.

The net effective radiative forcing ($ERF_{SW+LW}$) is simply the sum of $ERF_{SW}$ and $ERF_{LW}$:

$$ERF_{SW+LW} = \Delta(F + L) = ERF_{SW} + ERF_{LW} \approx ERF_{SW} + \Delta CRE_{LW}. \tag{7}$$

All the quantities mentioned in Eqs. (1)–(7) are calculated at TOA.

We also consider absorption by aerosols in the atmosphere ($AAA_{SW}$), defined as follows:

$$AAA_{SW} = (F - F_{clean}) - (F^{surface} - F_{clean}^{surface}) \tag{8}$$

where $F^{surface}$ is the net shortwave flux at the earth's surface, and $F_{clean}^{surface}$ is the clean-sky net shortwave flux at the earth's surface.

## 3 Results

To provide context for the discussion of the radiative effects, we first examine the aerosol mass budgets, lifetimes, and column burdens. We then focus on model output fields relating to different components of the ERF, taking each component in turn: the direct radiative effect, the cloud radiative effect, and the surface albedo radiative effect. When discussing each of these components, we also discuss related model fields: for example, in the section discussing the direct radiative effect we also consider other fields related to direct aerosol–radiation interactions.

### 3.1 Aerosol mass budgets and lifetimes

Tables 1–4 summarise the aerosol mass budgets for the year-2000 MARC simulation; Tables S1–S4 summarise the mass budgets for the year-1850 MARC simulation. Tables 3–8 of Liu et al. (2012) contain mass budgets for year-2000 MAM3 and MAM7 simulations. In this section, we focus primarily on the MARC mass budgets; Liu et al. (2012) discuss the MAM mass budgets.

For the MARC simulations, the majority of the sulfate aerosol mass exists in the accumulation-mode, with smaller amounts in MOS and MBS; very little exists in the nucleation and Aitken modes (Tables 1 and S1). Hydrometeors – both cloud droplets and precipitation – also contain dissolved sulfate, due to aqueous-phase oxidation of sulfur dioxide and scavenging of sulfate. In fact, evaporation of hydrometeors, which adds sulfate to the accumulation-mode, is by far the largest source of sulfate aerosol. The largest sinks of sulfate aerosol are nucleation scavenging by stratiform clouds and impaction scavenging. Scavenging does not necessarily imply permanent removal from the atmosphere: due to hydrometeor evaporation, the wet deposition rate (not diagnosed) is less than the sum of the scavenging rates. This rapid cloud-cycling of sulfate aerosol contributes to a short sulfate aerosol lifetime of 0.9 days, with accumulation-mode sulfate having an even shorter lifetime of 0.7 days. Due to the inclusion of cloud-cycling in the sources and sinks, the sulfate aerosol lifetime for MARC is not directly comparable to MAM: Liu et al. (2012), who reported a sulfate aerosol lifetime of approximately 4 days for MAM3 and MAM7, excluded cloud-cycling from the calculation. In the year-2000 MARC simulation, the lifetime of sulfate in MOS and MBS is approximately 4 days, a much longer lifetime than that of the rapidly cycled accumulation-mode sulfate. Interestingly, the lifetime of sulfate in MOS and MBS decreases from 6 days in the year-1850 simulation to 4 days in the year-2000 simulation. The increased availability of accumulation-mode sulfate aerosol in year-2000 appears to drive a large increase in the rate of coagulation on MOS and MBS, accelerating the growth of the MOS and MBS particles, likely decreasing the lifetime of the particles because larger particles are more likely to be removed through nucleation scavenging.

Organic carbon aerosol, including a large contribution from volatile organic compounds, is emitted into the pure OC mode in MARC (Tables 2 and S2). The majority of the pure OC mode aerosol is removed from the atmosphere by impaction scavenging. (Hydrometeor evaporation replenishes only sulfate aerosol in MARC, so scavenging acts as a permanent sink for the carbonaceous, dust, and sea salt aerosols.) However, some of the organic carbon aerosol in the pure

OC mode is transferred to the MOS mode by aging and coagulation. In contrast to the pure OC mode, which has a very low hygroscopicity, the largest sink for the MOS mode is nucleation scavenging by stratiform clouds: mixing the organic carbon aerosol with sulfate increases the hygroscopicity, allowing many of the MOS particles to be activated. However, despite the higher hygroscopicity, the organic carbon aerosol in the MOS mode has a slightly longer lifetime than the pure OC mode. As was the case for sulfate, the lifetime of organic carbon in the MOS mode decreases from approximately 6 days in the

year-1850 simulation to 4 days in the year-2000 simulation – as discussed above, this is likely due to increased availability of sulfate accelerating the growth of the MOS particles. When total organic carbon is considered, the organic carbon aerosol lifetime is approximately 5 days for both the year-1850 and year-2000 simulations. For MAM3 and MAM7, the primary organic matter aerosol lifetime is less than 5 days, while the secondary organic aerosol lifetime is 4 days (Liu et al., 2012). MARC's inclusion of a pure organic carbon mode with very low hygroscopicity likely contributes to the longer lifetime of

MARC's organic carbon aerosol compared with MAM's organic matter aerosol.

Black carbon aerosol is emitted into the pure BC mode in MARC (Tables 3 and S3). The majority of pure BC is removed from the atmosphere by impaction scavenging, although some is transferred to the MBS mode by aging. Based on the assumption of a core-shell model, MBS has the same hygroscopicity as sulfate, enabling nucleation scavenging by convective clouds to become the largest sink of black carbon aerosol in MBS, although impaction scavenging also remains a

major sink. BC in MBS has a slightly shorter lifetime than pure BC. The pure BC lifetime decreases from approximately 6 days in the year-1850 simulation to 4 days in the year-2000 simulation, primarily due to a substantially increased rate of aging. When total black carbon is considered, the black carbon aerosol lifetime is approximately 6 days in the year-1850 simulation and 5 days in the year-2000 simulation. The black carbon aerosol lifetime for MAM3 and MAM7 is approximately 4 days (Liu et al., 2012). MARC's inclusion of a pure black carbon mode with very low hygroscopicity likely

contributes to the longer lifetime of black carbon aerosol for MARC compared with MAM.

Sea salt aerosol emissions, which are dependent on simulated wind speed and sea surface temperature, are approximately 5.5 Pg yr$^{-1}$ in the MARC simulations (Tables 4 and S4); Liu et al. (2012) report slightly lower emissions of 5.0 Pg yr$^{-1}$ for MAM. For MARC and MAM7, dry deposition is the largest sink of sea salt aerosol; for MAM3, wet deposition and dry deposition are approximately equal sinks. The sea salt aerosol lifetime is approximately 0.6 days for

MARC and MAM7, and 0.8 days for MAM3.

Dust aerosol emissions, which are dependent on simulated wind speed and soil properties, are approximately 3.7 Pg yr$^{-1}$ in the year-2000 MARC simulation (Table 4); Liu et al. (2012) report lower emissions of 3.1 Pg yr$^{-1}$ for MAM3 and 2.9 Pg yr$^{-1}$ for MAM7. For MARC, impaction scavenging and dry deposition play approximately equal roles in removing dust aerosol from the atmosphere; for MAM, dry deposition dominates. The dust aerosol lifetime is approximately 4 days for

MARC and 3 days for MAM. The lifetime of dust in the year-1850 MARC simulation (Table S4) is very similar to that in the year-2000 simulation. However, dust emissions are slightly lower for the year-2000 simulation compared with the year-1850 simulation.

## 3.2 Aerosol column burdens

An aerosol column burden, also referred to as a loading, is the total mass of a given aerosol species in an atmospheric column. The advantage of column burdens is that they are relatively simple to understand, facilitating comparison between the different aerosol modules. However, when comparing the column burdens, it is important to remember that information about aerosol size distribution and aerosol mixing state is hidden. Information about the vertical distribution is also hidden, because the burdens are integrated throughout the atmospheric column.

**3.2.1 Total sulfate aerosol burden**

Figure 2a–c shows the total sulfate aerosol burden ($Burden_{SO4}$) for the year-2000 simulations. For all three aerosol modules, year-2000 $Burden_{SO4}$ is highest in the Northern Hemisphere subtropics and mid-latitudes, especially near source regions with high anthropogenic emissions of sulfur dioxide. Year-2000 $Burden_{SO4}$ is much lower in the Southern Hemisphere, especially over the remote Southern Ocean and Antarctica. In general, there is close agreement between MAM

and MARC over the Northern Hemisphere tropics and the Southern Hemisphere. However, over the Northern Hemisphere subtropics, mid-latitudes, and high-latitudes, year-2000 $Burden_{SO4}$ is generally lower for MARC compared with MAM3. Interestingly, over the Northern Hemisphere subtropics, the zonal means are very similar between MAM7 and MARC. The differences between MARC, MAM3, and MAM7 may be due to differences in sulfate aerosol lifetime. However, it is not possible to test this conclusively: as pointed out in Section 3.1, the sulfate aerosol lifetimes diagnosed for MARC should not

be directly compared to those diagnosed for MAM, because cloud-cycling contributes to the sources and sinks of sulfate aerosol in MARC.

Figure 2d–f shows $\Delta Burden_{SO4}$, the 2000-1850 difference in $Burden_{SO4}$. Both MAM3 and MARC produce widespread positive values of $\Delta Burden_{SO4}$ across the Northern Hemisphere and also across South America, Africa, and Oceania. For both MAM and MARC, global mean $\Delta Burden_{SO4}$ accounts for more than half of global mean year-2000

$Burden_{SO4}$, indicating that anthropogenic sulfur emissions are responsible for more than half of the global burden of sulfate aerosol.

**3.2.2 Total organic matter and total organic carbon aerosol burdens**

Figure 3a–c shows the total organic matter aerosol burden ($Burden_{OM}$) for MAM and the total organic carbon aerosol burden ($Burden_{OC}$) for MARC for the year-2000 simulations. Due to different handling of organic carbon aerosol, MAM's

$Burden_{OM}$ is not directly equivalent to MARC's $Burden_{OC}$.

For both MAM3 and MARC, year-2000 $Burden_{OM}$ and $Burden_{OC}$ peak in the tropics, especially sub-Saharan Africa and South America, due to emissions from wildfires. The impact of anthropogenic emissions of organic carbon aerosol is evident over South Asia and East Asia. Biogenic emissions of isoprene and monoterpene also contribute to $Burden_{OM}$ and $Burden_{OC}$. Although the fields are not directly equivalent, it is interesting to note that MARC's $Burden_{OC}$

is less than MAM's $Burden_{OM}$ at latitudes with substantial emissions sources but greater than MAM's $Burden_{OM}$ at high-latitudes far from substantial emissions sources: this is consistent with the longer lifetime of MARC's organic carbon aerosol compared with MAM's organic matter and secondary organic aerosol.

Over the major emissions regions of organic carbon aerosol, MAM3 and MARC generally produce positive values of $\Delta Burden_{OM}$ and $\Delta Burden_{OC}$, the 2000-1850 differences in $Burden_{OM}$ and $Burden_{OC}$ (Fig. 3d–f). However, negative

values of $\Delta Burden_{OM}$ and $\Delta Burden_{OC}$ are found over North America. These 2000-1850 differences arise due to changes in both wildfire emissions and anthropogenic emissions of organic carbon aerosol between year-1850 and year-2000. Although emissions of some volatile organic compounds do change between year-1850 and year-2000, emissions of isoprene and monoterpene remain unchanged so these species are unlikely to contribute to $\Delta Burden_{OM}$ and $\Delta Burden_{OC}$.

### 3.2.3 Total black carbon aerosol burden

Figure 4a–c shows the total black carbon aerosol burden ($Burden_{BC}$) for the year-2000 simulations. For both MAM3 and MARC, year-2000 $Burden_{BC}$ is high over sub-Saharan Africa and South America due to large emissions of black carbon aerosol from wildfires. However, the peak in zonal mean year-2000 $Burden_{BC}$ occurs in the Northern Hemisphere subtropics and mid-latitudes, due to anthropogenic emissions of black carbon aerosol over East Asia, South Asia, and Europe. Year-2000 $Burden_{BC}$ for MARC is generally higher than that for MAM, especially over remote regions far away

from sources. This is consistent with the longer lifetime of black carbon aerosol lifetime for MARC compared with MAM, likely due to the low hygroscopicity of MARC's pure black carbon aerosol mode.

MAM3 and MARC produce similar increases in $Burden_{BC}$ between year-1850 and year-2000, as indicated by positive values of $\Delta Burden_{BC}$ (Fig. 4d–f). For MARC, positive values of $\Delta Burden_{BC}$ are found over even remote ocean regions, consistent with a longer black carbon lifetime for MARC compared with MAM3. Were it not for the decrease in

MARC's black carbon aerosol lifetime between year-1850 (Table S3) and year-2000 (Table 3), $\Delta Burden_{BC}$ would be even larger.

### 3.2.4 Total sea salt aerosol burden

Figure 5a–c shows the total sea salt aerosol burden ($Burden_{salt}$) for the year-2000 simulations. For both MAM3 and MARC, year-2000 $Burden_{salt}$ is highest over ocean areas with strong surface wind speeds (Fig. S9b, c). Over land, year-

2000 $Burden_{salt}$ is very low, due to the short lifetime of sea salt aerosol. Year-2000 $Burden_{salt}$ is very similar between

MAM3 and MARC. However, sea salt emissions and lifetime do vary slightly between MARC, MAM3, and MAM7, as noted in Section 3.1.

For both MAM3 and MARC, $\Delta Burden_{salt}$, the 2000-1850 difference in $Burden_{salt}$ (Fig. 5e, f), appears to be positively correlated with the aerosol-induced 2000-1850 difference in surface wind speeds (Fig. S9e, f). Changes in
precipitation rate (Fig. S8e, f) likely also influence $Burden_{salt}$, because precipitation efficiently removes sea salt aerosol from the atmosphere. However, it should be noted that the aerosol-induced 2000-1850 differences in $Burden_{salt}$, surface wind speed, and precipitation rate are both relatively small and often statistically insignificant across most of the world. Hence these 2000-1850 differences may be due primarily to internal variability. If an interactive dynamical ocean were to be used, allowing SSTs to respond to the anthropogenic aerosol ERF, it is likely that we would find much larger aerosol-
induced 2000-1850 differences in surface wind speed, precipitation rate, and $Burden_{salt}$.

### 3.2.5 Total dust aerosol burden

Figure 6a–c shows the total dust aerosol burden ($Burden_{dust}$) for the year-2000 simulations. Dust emission primarily occurs over desert areas, especially the Sahara Desert, so year-2000 $Burden_{dust}$ is highest directly over and downwind of these desert source regions. Year-2000 $Burden_{dust}$ is much larger for MARC, which follows Albani et al., (2014), compared with
MAM. The largest differences between MAM3 and MARC appear to occur directly over the desert source regions, suggesting that differences in dust emission drive the differences in year-2000 $Burden_{dust}$ – if this is the case, dust emission is far higher for MARC compared with MAM over the Sahara, Middle East, and East Asian deserts, while the opposite may be true over southern Africa and Australia. As mentioned in Section 3.1, global dust emissions are indeed higher for MARC compared with MAM. Differences in the lifetime of dust aerosol also contribute to the differences in year-2000 $Burden_{dust}$
between MAM and MARC: the dust aerosol lifetime is 3 days for MAM and 4 days for MARC.

$\Delta Burden_{dust}$, the aerosol-induced 2000-1850 difference in $Burden_{dust}$ (Fig. 6d–f), reveals that $Burden_{dust}$ decreases between the year-1850 and year-2000 simulations, especially over the Sahara Desert. Both MAM3 and MARC produce a similar zonal mean decrease in $Burden_{dust}$. The reasons for the aerosol-induced 2000-1850 changes in $Burden_{dust}$ are unclear, although changes in surface wind speed (Fig. S9d–f), influencing emission, likely contribute: for
MARC, global dust emissions are slightly lower in the year-2000 simulation (Table 4) compared to the year-1850 simulation (Table S4). As we noted above when discussing the sea salt burden, if an interactive dynamical ocean were to be used, it is likely that we would find much larger aerosol-induced 2000-1850 differences in surface wind speed, precipitation rate, and $Burden_{dust}$.

### 3.3 Aerosol–radiation interactions and the direct radiative effect

### 3.3.1 Aerosol optical depth

Aerosols scatter and absorb shortwave radiation, leading to extinction of incoming solar radiation. Before considering the direct radiative effect, we first look at aerosol optical depth ($AOD$), a measure of the total extinction due to aerosols in an atmospheric column.

Figure 7a–c shows $AOD$ for the year-2000 simulations. For both MAM and MARC, zonal mean year-2000 $AOD$ peaks in the Northern Hemisphere subtropics, driven by emission of dust from deserts, especially the Sahara Desert. Over other regions, both anthropogenic aerosol emissions and natural aerosol emissions, including emissions of sea salt, contribute to year-2000 $AOD$. The year-2000 $AOD$ values for MARC are often much lower than those for MAM3, especially over subtropical ocean regions. Rothenberg et al. (2018) have also previously noted that the $AOD$ for MARC is generally lower than that retrieved from the MODerate Resolution Imaging Spectroradiometer (MODIS; Collection 5.1); but it should be noted that differences in spatial-temporal sampling (Schutgens et al., 2017, 2016) have not been accounted for, and satellite-based retrievals of $AOD$ are not error-free.

The differences between the aerosol burdens for MAM and MARC, discussed above, are insufficient to explain the differences in year-2000 $AOD$, especially the large difference in the Southern Hemisphere subtropics (where the burdens are similar between MAM and MARC). Hence it is likely that differences in the aerosol optical properties between MARC and MAM are responsible for the fact that MARC generally produces lower values of $AOD$. In particular, the large difference in year-2000 $AOD$ between MAM3 and MARC over the subtropical ocean, where $Burden_{salt}$ is large, is likely due to differences in the optical properties of sea salt aerosol.

Positive 2000-1850 differences in $Burden_{SO4}$, $Burden_{OM}/Burden_{OC}$, and $Burden_{BC}$, discussed above, drive positive values of $\Delta AOD$, the 2000-1850 difference in $AOD$ (Fig. 7d–f). As was the case for year-2000 $AOD$, $\Delta AOD$ produced by MARC is generally much lower than $\Delta AOD$ produced by MAM3.

### 3.3.2 Direct radiative effect

Figure 8a–c shows the direct radiative effect ($DRE_{SW}$) for the year-2000 simulations. $DRE_{SW}$ reveals the influence of direct interactions between radiation and aerosols on the net shortwave flux at TOA (Eq. (3)). Aerosols that scatter shortwave radiation efficiently, such as sulfate, generally contribute to negative values of $DRE_{SW}$, indicating a cooling effect on the climate system; aerosols that absorb shortwave radiation, such as black carbon, generally contribute to positive values of $DRE_{SW}$, indicating a warming effect on the climate system. Other factors, such as the presence of clouds, the vertical distribution of aerosols relative to clouds, and the albedo of the earth's surface, also play a role in determining $DRE_{SW}$ (Stier et al., 2007). Due to these factors – especially the differing impact of scattering and absorbing aerosols and variations in the albedo of the earth's surface – large values of $AOD$ may not necessarily correspond to large values of $DRE_{SW}$. Having said that, for both MAM3 and MARC, the regional distribution of year-2000 $DRE_{SW}$ shares some similarities with that of year-

2000 $AOD$. Over dark ocean surfaces in the subtropics, scattering by aerosols drives negative values of year-2000 $DRE_{SW}$. The impact of dust on year-2000 $DRE_{SW}$ differs between MAM3 and MARC, likely due to differing optical properties: for MAM3, absorption by dust drives positive values over the bright surface of the Sahara Desert, while little radiative impact is evident downwind over the dark surface of the tropical Atlantic Ocean; for MARC, scattering by dust drives negative values over the tropical Atlantic Ocean, while little radiative impact is evident over the Sahara Desert. The impact of black carbon aerosol on year-2000 $DRE_{SW}$ also differs between MAM3 and MARC: for MAM3, absorption by black carbon aerosol drives positive values of $DRE_{SW}$ over the South Atlantic stratocumulus deck near southern Africa; for MARC, negative values of $DRE_{SW}$ are found over the stratocumulus deck, suggesting weaker absorption by black carbon aerosol compared with MAM3. The differing absorption is likely due to the differing representations of aerosol mixing state and associated optical properties: the majority of MARC's black carbon aerosol is found in the pure BC mode, whereas MAM's black carbon aerosol is internally mixed with other species, likely leading to stronger absorption for MAM compared with MARC.

For MAM, $\Delta DRE_{SW}$, the 2000-1850 difference in $DRE_{SW}$, is relatively weak at all latitudes (Fig. 8d, e), with a global mean of only $-0.02 \pm 0.01$ W m$^{-2}$ for MAM3 and $-0.00 \pm 0.01$ W m$^{-2}$ for MAM7 (Table 5), due to the cooling effect of anthropogenic sulfur emissions being offset by the warming effect of increased black carbon aerosol emissions (Ghan et al., 2012). In contrast, for MARC, $\Delta DRE_{SW}$ reveals a relatively strong cooling effect across much of the Northern Hemisphere (Fig. 8d, f), especially near anthropogenic sources of sulfur emissions, leading to a global mean $\Delta DRE_{SW}$ of $-0.17 \pm 0.01$ W m$^{-2}$.

### 3.3.3 Absorption by aerosols in the atmosphere

Figure 9a–c shows the absorption of shortwave radiation by aerosols in the atmosphere ($AAA_{SW}$; Eq. (8)) for the year-2000 simulations. Consideration of $AAA_{SW}$, which reveals heating of the atmosphere by aerosols, complements consideration of $DRE_{SW}$. For example, over the Sahara Desert, we noted above that the dust aerosol in MARC exerts only a weak direct radiative effect at TOA (Fig. 8c); however, Fig. 9c reveals that the dust aerosol in MARC leads to strong heating of the atmosphere. For both MAM and MARC, year-2000 $AAA_{SW}$ is largest near emission sources of dust, especially over the Sahara Desert where year-2000 $Burden_{dust}$ is particularly high, showing that dust is the primary driver of year-2000 $AAA_{SW}$. Further away from the dust emission source regions, year-2000 $AAA_{SW}$ is spatially correlated with year-2000 $Burden_{BC}$, showing that black carbon aerosol also contributes to year-2000 $AAA_{SW}$. Despite the fact that year-2000 $Burden_{dust}$ and $Burden_{BC}$ are larger for MARC compared with MAM, year-2000 $AAA_{SW}$ is generally weaker for MARC compared with MAM: the weaker absorption for MARC is likely due to differences in the aerosol optical properties, associated with different handling of dust and black carbon aerosol mixing state between MAM and MARC.

$\Delta AAA_{SW}$, the 2000-1850 difference in $AAA_{SW}$ (Fig. 9d–f), generally follows the same regional distribution as $\Delta Burden_{BC}$, showing that changes in emissions of black carbon aerosol dominate $\Delta AAA_{SW}$. Although dust dominates year-

2000 $AAA_{SW}$, changes in dust emission exert only a relatively small influence on $\Delta AAA_{SW}$. As with year-2000 $AAA_{SW}$, $\Delta AAA_{SW}$ is generally weaker for MARC compared with MAM3.

The absorption of shortwave radiation by aerosols can drive rapid adjustments of the atmosphere, such as changes to atmospheric stability and humidity, influencing clouds (Stjern et al., 2017). Such "semidirect" effects may contribute to the cloud radiative effects discussed below. However, Ghan et al. (2012) found both the shortwave and longwave semidirect radiative effects to be statistically insignificant for MAM3 and MAM7; we expect the same to apply for MARC, because the default CAM5.3 cloud microphysics scheme is also used. Cloud microphysical effects dominate the cloud radiative effects.

## 3.4 Aerosol–cloud interactions and the cloud radiative effects

### 3.4.1 Cloud condensation nuclei concentration

Many aerosol particles have the potential to become the cloud condensation nuclei (CCN) on which water vapour condenses to form cloud droplets. Figure 10a–c shows the CCN concentration at a fixed supersaturation of 0.1% ($CCN_{conc}$) in the lower troposphere for the year-2000 simulations. Corresponding results showing year-2000 $CCN_{conc}$ near the surface and in the mid-troposphere are shown in Figs. S1a–c and S2a–c of the Supplement. Looking at year-2000 $CCN_{conc}$ across these different vertical levels, we make two initial observations: first, for both MAM and MARC, year-2000 $CCN_{conc}$ is generally higher in the Northern Hemisphere; second, year-2000 $CCN_{conc}$ is generally much lower for MARC compared with MAM.

When we look in more detail at the regional distribution of year-2000 $CCN_{conc}$ for MAM3, and compare this to the column burden results, we notice that locations with high $CCN_{conc}$ have either high $Burden_{SO4}$ or high $Burden_{OM}$. This suggests that, for MAM3, the organic matter aerosol – internally-mixed with other species with high hygroscopicity – contributes to efficient CCN, consistent with two previous MAM3-based studies that found that organic carbon emissions from wildfires can exert a strong influence on clouds (Grandey et al., 2016a; Jiang et al., 2016).

In contrast, for MARC, the regional distribution of year-2000 $CCN_{conc}$ closely resembles that of $Burden_{SO4}$ but does not resemble that of $Burden_{OC}$. This suggests that, for MARC, the organic carbon aerosol – much of which remains in a pure organic carbon aerosol mode with very low hygroscopicity – is not an efficient source of CCN.

If we look at the results for $\Delta CCN_{conc}$, the 2000-1850 difference in $CCN_{conc}$ (Figs. 10d–f, S1d–f, and S2d–f), similar deductions about sulfate aerosol and organic carbon aerosol can be made as were made above. For MAM3, the regional distribution of $\Delta CCN_{conc}$ reveals that changes in the availability of CCN are associated with both $\Delta Burden_{SO4}$ and $\Delta Burden_{OM}$. For MARC, the regional distribution of $\Delta CCN_{conc}$ is associated with $\Delta Burden_{SO4}$, but is not closely associated with $\Delta Burden_{OC}$.

For both MAM and MARC, $\Delta CCN_{conc}$ is generally positive, revealing increasing availability of CCN between year-1850 and year-2000. The absolute increase is smaller for MARC than for MAM. However, the percentage increase is larger for MARC than for MAM.

It is important to note that these $CCN_{conc}$ results are for a fixed supersaturation of 0.1%; but as pointed out by Rothenberg et al. "*all* aerosol [particles] are potentially CCN, given an updraft sufficient enough in strength to drive a high-enough supersaturation such that they grow large enough to activate" (Rothenberg et al., 2018). Furthermore, the number of CCN that are actually activated is influenced by competition for water vapour among various types of aerosol particles,
which depends on the details of the aerosol population including size distribution and mixing state. When aerosol particles with a lower hygroscopicity rise alongside aerosol particles with a higher hygroscopicity in a rising air parcel, the latter would normally be activated first at a supersaturation that is much lower than the one required for the former to become activated; the consequent condensation of water vapour to support the diffusive growth of the newly formed cloud particles would effectively lower the saturation level of the air parcel and further reduce the chance for the lower hygroscopicity
aerosol particles to be activated (Rothenberg and Wang, 2016, 2017). In other words, $CCN_{conc}$ at a fixed supersaturation is not necessarily a good indicator of the number of CCN that are actually activated, because activation depends on specific environmental conditions and the details of the aerosol population present.

Differences in the representation of aerosol mixing state and hygroscopicity may lead to large differences in aerosol activation spectra. In an aerosol model such as MAM that includes only internally-mixed modes, the hygroscopicity of a
given mode is derived by volume weighting through all the included aerosol species and is therefore not very sensitive to changes in the chemical composition of the mode. In contrast, MARC explicitly handles mixing state and thus hygroscopicity of each individual type of aerosol: for example, the hygroscopicity of the pure OC and pure BC modes is very low, the hygroscopicity of the MOS mode depends on the internal mixing state of organic carbon and sulfate (assuming homogeneous mixing), and the hygroscopicity of the MBS mode is as high as that of pure sulfate (assuming a core-shell
model) (Rothenberg et al., 2018).

### 3.4.2 Column-integrated cloud droplet number concentration

The availability of CCN influences cloud microphysics via the formation of cloud droplets. Figure 11a–c shows column-integrated cloud droplet number concentration ($CDNC_{column}$) for the year-2000 simulations. For MAM, year-2000 $CDNC_{column}$ is generally higher in the Northern Hemisphere, with very high values occurring over regions with abundant
sulfate aerosol or organic carbon aerosol providing abundant CCN. In contrast, for MARC there is no strong inter-hemispheric asymmetry in year-2000 $CDNC_{column}$: there appears to be no influence from organic carbon aerosol, consistent with the $CCN_{conc}$ results discussed above; and the influence of sulfate aerosol appears weaker than for MAM.

No global observations of $CDNC_{column}$ exist. However, satellite-based estimates of cloud-top cloud droplet number concentration ($CDNC_{top}$) can be derived using the adiabatic assumption, although the uncertainties are large. MARC tends
to underestimate $CDNC_{top}$ compared to MODIS-derived estimates (Rothenberg et al., 2018).

When we look at $\Delta CDNC_{column}$, the 2000-1850 difference in $CDNC_{column}$ (Fig. 11d–f), we see that anthropogenic emissions generally drive increases in $CDNC_{column}$, as expected. The absolute increase is smaller for MARC than for MAM.

However, the percentage increase is comparable between MARC and MAM. As was the case for $\Delta CCN_{\text{conc}}$, for MAM3, the regional distribution of $\Delta CDNC_{\text{column}}$ appears to be associated with both $\Delta Burden_{\text{SO4}}$ and $\Delta Burden_{\text{OM}}$, whereas for MARC, the regional distribution of $\Delta CDNC_{\text{column}}$ is associated with $\Delta Burden_{\text{SO4}}$ but is not closely associated with $\Delta Burden_{\text{OC}}$.

### 3.4.3 Grid-box cloud liquid and cloud ice water paths

In addition to influencing cloud microphysical properties (such as cloud droplet number concentration), the availability of CCN and ice nuclei influence cloud macrophysical properties (such as cloud water path). Figure 12a–c shows grid-box cloud liquid water path ($WP_{\text{liquid}}$) for the year-2000 simulations. Year-2000 $WP_{\text{liquid}}$ is highest in the tropics and mid-latitudes. The regional distribution of year-2000 $WP_{\text{liquid}}$ is similar to that of total cloud fractional coverage (Fig. S4a–c). The regional distribution of year-2000 $WP_{\text{liquid}}$ for MARC is very similar to that for MAM. However, compared with MAM, MARC produces slightly higher year-2000 $WP_{\text{liquid}}$ in the Southern Hemisphere mid-latitudes, the Southern Hemisphere subtropics, and the Arctic.

Figure 13a–c shows grid-box cloud ice water path ($WP_{\text{ice}}$) for the year-2000 simulations. As with $WP_{\text{liquid}}$, year-2000 $WP_{\text{ice}}$ is highest in the tropics and mid-latitudes. The regional distribution of year-2000 $WP_{\text{ice}}$ is similar to that of high-level cloud fractional coverage (Fig. S7a–c), and is similar between MAM and MARC. However, year-2000 $WP_{\text{ice}}$ is consistently lower for MARC than for MAM. Although MARC and MAM3 are coupled to the same ice and mixed-phase cloud microphysics scheme (Gettelman et al., 2010; Liu et al., 2007), differences in the availability of ice nuclei can arise due to differences in dust and sulfate number concentrations and size distributions. The uncertainties associated with ice nucleation are very large (Garimella et al., 2018).

The 2000-1850 differences in $WP_{\text{liquid}}$ and $WP_{\text{ice}}$ are shown in Figs. 12d–f and 13d–f. MAM3 produces large increases in $WP_{\text{liquid}}$ over Europe, East Asia, Southeast Asia, South Asia, parts of Africa, and northern South America – the regional distribution of $\Delta WP_{\text{liquid}}$ is similar to the regional distributions of $\Delta CCN_{\text{conc}}$ and $\Delta CDNC_{\text{column}}$. MARC produces large increases in $WP_{\text{liquid}}$ over the same regions, and additionally over Australia and North America. $\Delta WP_{\text{liquid}}$ is often larger for MARC than for MAM3, especially over the Northern Hemisphere mid-latitudes. For MARC, in comparison with MAM3, the relatively strong $\Delta WP_{\text{liquid}}$ response is consistent with the relatively strong $\Delta CCN_{\text{conc}}$ percentage change.

Globally, for both MAM3 and MARC, the $\Delta WP_{\text{ice}}$ response is relatively weak (Fig. 13d–f). However, relatively large values of $\Delta WP_{\text{ice}}$, both positive and negative, are found regionally. This regional response differs between MAM3 and MARC. For both MAM3 and MARC, it appears that decreases in $WP_{\text{ice}}$ correspond to increases in $Burden_{\text{OM}}$ and $Burden_{\text{OC}}$ (Fig. 3e, f); but this relationship is likely spurious, because organic carbon aerosol does not directly influence ice processes in either aerosol module.

### 3.4.4 Shortwave cloud radiative effect

Figure 14a–c shows the clean-sky shortwave cloud radiative effect ($CRE_{SW}$; Eq. (4)) for the year-2000 simulations. Clouds scatter much of the incoming solar radiation, exerting a strong cooling effect on the climate system. Globally, year-2000 $CRE_{SW}$ is $-53.9$ W m$^{-2}$ for MAM3 and $-52.2$ W m$^{-2}$ for MARC. For both MAM and MARC, $CRE_{SW}$ is strongest in the tropics and mid-latitudes. The regional distribution of year-2000 $CRE_{SW}$ is negatively correlated with $WP_{liquid}$ and $WP_{total}$ (the total cloud water path; Fig. S3): large values of $WP_{liquid}$ and $WP_{total}$ correspond to a strong cooling effect.

The same applies to $\Delta CRE_{SW}$, the 2000-1850 difference in $CRE_{SW}$ (Fig. 14d–f), which is negatively correlated with $\Delta WP_{liquid}$ and $\Delta WP_{total}$: increases in $WP_{liquid}$ and $WP_{total}$ drive a stronger shortwave cloud cooling effect. For both MAM3 and MARC, the cooling effect of $\Delta CRE_{SW}$ is strongest in the Northern Hemisphere, particularly regions with high anthropogenic sulfur emissions, especially East Asia, Southeast Asia, and South Asia. Compared with MAM, MARC produces a slightly stronger $\Delta CRE_{SW}$ response in the Northern Hemisphere mid-latitudes and a slightly weaker $\Delta CRE_{SW}$ response in the Northern Hemisphere sub-tropics. Another difference between MAM3 and MARC is the land-ocean contrast: compared with MAM3, MARC often produces a slightly stronger $\Delta CRE_{SW}$ response over land but a weaker $\Delta CRE_{SW}$ response over ocean. In particular, MARC produces a weaker $\Delta CRE_{SW}$ response over the stratocumulus decks near South America and southern Africa, likely because organic carbon aerosol is not an efficient source of CCN in MARC.

When globally averaged, the global mean $\Delta CRE_{SW}$ for MARC ($-2.17 \pm 0.04$ W m$^{-2}$) is stronger than that for MAM3 ($-2.09 \pm 0.04$ W m$^{-2}$) and MAM7 ($-2.05 \pm 0.04$ W m$^{-2}$). The stronger $\Delta CRE_{SW}$ response for MARC is consistent with the larger $\Delta CCN_{conc}$ percentage change for MARC compared with MAM.

### 3.4.5 Longwave cloud radiative effect

The cooling effect of $CRE_{SW}$ is partially offset by the warming effect of $CRE_{LW}$, the longwave cloud radiative effect that arises due to absorption of longwave thermal infrared radiation (Eq. (6)). Figure 15a–c shows $CRE_{LW}$ for the year-2000 simulations. Globally, year-2000 $CRE_{LW}$ is $+24.1$ W m$^{-2}$ for MAM3 and $+22.2$ W m$^{-2}$ for MARC. As with the shortwave cooling effect, the longwave warming effect is strongest in the tropics and mid-latitudes, for both MAM and MARC. The regional distribution of year-2000 $CRE_{LW}$ is positively correlated with $WP_{ice}$ (Fig. 13a–c) and high-level cloud fraction (Fig. S7a–c) – high-level ice cloud drives the longwave warming effect.

The same is true for $\Delta CRE_{LW}$, the 2000-1850 difference in $CRE_{LW}$ (Fig. 15d–f): changes in high-level ice cloud cover drive changes in the longwave cloud warming effect. For both MAM3 and MARC, $\Delta CRE_{LW}$ is positive over much of Southeast Asia, South Asia, the Indian Ocean, the Atlantic Ocean, and the Pacific Ocean; $\Delta CRE_{LW}$ is negative over much of Africa and parts of South America. MAM3 produces a global mean $\Delta CRE_{LW}$ of $+0.54 \pm 0.02$ W m$^{-2}$ and MAM7 produces a global mean $\Delta CRE_{LW}$ of $+0.53 \pm 0.02$ W m$^{-2}$, while MARC produces a stronger global mean of $+0.66 \pm 0.01$ W m$^{-2}$. Hence $\Delta CRE_{LW}$ offsets approximately one quarter to one third of the $\Delta CRE_{SW}$ cooling effect.

### 3.5 The surface albedo radiative effect

In addition to interacting with radiation both directly and indirectly via clouds, aerosols can influence the earth's radiative energy balance via changes to the surface albedo. The surface albedo radiative effect ($\Delta SRE_{SW}$; Eq. (5)), "includes effects of both changes in snow albedo due to deposition of absorbing aerosol, and changes in snow cover induced by deposition and by the other aerosol forcing mechanisms" (Ghan, 2013). For both MAM3 and MARC, deposition of absorbing aerosol is enabled via the coupling between CAM5 and the land scheme in CESM; and "other aerosol forcing mechanisms" include aerosol-induced changes in precipitation rate. Aerosol-induced changes in column water vapor can also influence the calculation of $\Delta SRE_{SW}$, because $F_{clean,clear}$ is sensitive to near-infrared absorption by water vapour; but the contribution from such changes in column water vapour is small. ($\Delta SRE_{SW}$ does not include changes in land use, because the only difference between the year-1850 and year-2000 simulations is the aerosol emissions.)

Figure 16 shows $\Delta SRE_{SW}$, the aerosol-induced 2000-1850 surface albedo radiative effect. In the Arctic and high-latitude land regions of the Northern Hemisphere, $\Delta SRE_{SW}$ can be relatively large. MAM3 produces a mixture of positive and negative $\Delta SRE_{SW}$ values, averaging out to zero globally ($+0.00 \pm 0.02$ W m$^{-2}$). However, MARC tends to produce mainly negative $\Delta SRE_{SW}$ values, averaging out to a global mean of $-0.10 \pm 0.02$ W m$^{-2}$.

The $\Delta SRE_{SW}$ response is associated with aerosol-induced 2000-1850 changes in snow cover over both land and sea-ice (Fig. S10d–f): increases in snow cover lead to negative $\Delta SRE_{SW}$ values, while decreases in snow cover lead to positive $\Delta SRE_{SW}$ values. Changes in snow rate (Fig. S11d–f) likely play a major role, explaining much of the snow cover response. Changes in black carbon deposition (Fig. S12d–f), contributing to changes in the mass of black carbon in the top layer of snow (Fig. S13d–f), may also play a role. The mass of black carbon in the top layer of snow is much lower for MARC compared with MAM3 (Fig. S13a–c), likely due to differences in dry deposition of black carbon: the rate of dry deposition of black carbon aerosol is 0.42 Tg year$^{-1}$ in MARC (Table 3), much lower than the rate of 1.27 Tg year$^{-1}$ in MAM3 (Liu et al., 2012). The aerosol-induced 2000-1850 difference in the mass of black carbon in the top layer of snow is also much lower for MARC compared with MAM3 (Fig. S13d–f).

### 3.6 Net effective radiative forcing

The net effective radiative forcing ($ERF_{SW+LW}$) – the 2000-1850 difference in the net radiative flux at TOA (Eq. (7)) – is effectively the sum of the radiative effect components we discussed above. Figure 17 shows $ERF_{SW+LW}$; Table 5 summarises the global mean contribution from the different radiative effect components.

In general, the cloud shortwave component, $\Delta CRE_{SW}$, dominates, resulting in negative values of $ERF_{SW+LW}$ across much of the world. In particular, strongly negative values of $ERF_{SW+LW}$, indicating a large cooling effect, are found near regions with substantial anthropogenic sulfur emissions. The cooling effect is far stronger in the Northern Hemisphere than it is in the Southern Hemisphere. If coupled atmosphere–ocean simulations were to be performed, allowing SSTs to

respond, the large inter-hemispheric difference in $ERF_{SW+LW}$ would likely impact inter-hemispheric temperature gradients and hence rainfall patterns (Chiang and Friedman, 2012; Grandey et al., 2016b; Wang, 2015).

Across much of the globe, the net cooling effect of $ERF_{SW+LW}$ produced by MARC is similar to that produced by MAM. However, in the mid-latitudes, MARC produces a stronger net cooling effect, especially over North America, Europe, and northern Asia. Another difference is that MARC appears to exert more widespread cooling over land than MAM3 does, while the opposite appears to be the case over ocean. These differences in the regional distribution of $ERF_{SW+LW}$ are largely due to differences in the regional distribution of $\Delta CRE_{SW}$. As mentioned in the previous paragraph, rainfall patterns are sensitive to changes in surface temperature gradients. Therefore, if SSTs were allowed to respond to the forcing, the differences in the regional distribution of $ERF_{SW+LW}$ between MARC and MAM may drive differences in rainfall patterns.

When averaged globally, MAM3 produces a global mean $ERF_{SW+LW}$ of $-1.57 \pm 0.04$ W m$^{-2}$ and MAM7 produces a global mean $ERF_{SW+LW}$ of $-1.53 \pm 0.04$ W m$^{-2}$; MARC produces a stronger global mean $ERF_{SW+LW}$ of $-1.79 \pm 0.03$ W m$^{-2}$. The $ERF_{SW+LW}$ produced by CAM5.3-MARC-ARG is particularly strong compared with many other global climate models (Shindell et al., 2013).

## 4 Summary and conclusions

The specific representation of aerosols in global climate models, especially the representation of aerosol mixing state, has important implications for aerosol hygroscopicity, aerosol lifetime, aerosol column burdens, aerosol optical properties, and cloud condensation nuclei (CCN) availability. For example, in addition to internally-mixed modes, MARC also includes a pure organic carbon aerosol mode and a pure black carbon aerosol mode both of which have very low hygroscopicity. The low hygroscopicity of these pure organic carbon and pure black carbon modes leads to increased lifetimes of total organic carbon aerosol and total black carbon aerosol, influencing aerosol column burdens. Furthermore, the representation of aerosol mixing state, and the associated implications for hygroscopicity, strongly influences the ability of the aerosol particles to act as CCN. For example, MARC's organic carbon aerosol is not an efficient source of CCN, but MAM's organic matter aerosol – internally-mixed with other species with high hygroscopicity – is an efficient source of CCN.

We have demonstrated that changing the aerosol module in CAM5.3 influences both the direct and indirect radiative effects of aerosols. Standard CAM5.3, which uses the MAM3 aerosol module, produces a global mean net ERF of $-1.57 \pm 0.04$ W m$^{-2}$ associated with the 2000-1850 difference in aerosol (including aerosol precursor) emissions; CAM5.3-MARC-ARG, which uses the MARC aerosol module, produces a stronger global mean net ERF of $-1.79 \pm 0.03$ W m$^{-2}$, a particularly strong cooling effect compared with other climate models (Shindell et al., 2013). For MARC compared with MAM, the stronger global mean net ERF can be attributed to stronger cooling via the direct radiative effect, the shortwave cloud radiative effect, and the surface albedo radiative effect, as summarised below. Furthermore, differences in the regional distribution of the shortwave cloud radiative effect drive differences in the regional distribution of net ERF.

By analysing the individual components of the net ERF, we have demonstrated that:

1. The global mean 2000-1850 direct radiative effect produced by MAM ($-0.02 \pm 0.01$ W m$^{-2}$ for MAM3; $-0.00 \pm 0.01$ W m$^{-2}$ for MAM7) is close to zero due to the warming effect of black carbon aerosol opposing the cooling effect of sulfate aerosol and organic carbon aerosol. In contrast, the 2000-1850 direct radiative effect produced by MARC is $-0.17 \pm 0.01$ W m$^{-2}$, with the cooling effect of sulfate aerosol being larger than the warming effect of black carbon aerosol.

2. The global mean 2000-1850 shortwave cloud radiative effect produced by MARC ($-2.17 \pm 0.04$ W m$^{-2}$) is stronger than that produced by MAM ($-2.09 \pm 0.04$ W m$^{-2}$ for MAM3; $-2.05 \pm 0.04$ W m$^{-2}$ for MAM7). Furthermore, the regional distribution differs: for MAM3, the cooling peaks in the Northern Hemisphere subtropics; while for MARC, the cooling peaks in the Northern Hemisphere mid-latitudes. The land-ocean contrast also differs: compared with MAM3, MARC often produces stronger cooling over land but weaker cooling over ocean. For both MAM3 and MARC, the 2000-1850 shortwave cloud radiative effect is closely associated with changes in liquid water path.

3. The global mean 2000-1850 longwave cloud radiative effect produced by MARC ($+0.66 \pm 0.01$ W m$^{-2}$) is stronger than that produced by MAM ($+0.54 \pm 0.02$ W m$^{-2}$ for MAM3; $+0.53 \pm 0.02$ W m$^{-2}$ for MAM7). For both MAM3 and MARC, the 2000-1850 longwave cloud radiative effect is closely associated with changes in ice water path and high cloud cover.

4. The global mean 2000-1850 surface albedo radiative effect produced by MARC ($-0.10 \pm 0.02$ W m$^{-2}$) is also stronger than that produced by MAM ($+0.00 \pm 0.02$ W m$^{-2}$ for MAM3; $-0.02 \pm 0.02$ W m$^{-2}$ for MAM7). The 2000-1850 surface albedo radiative effect is associated with changes in snow cover.

If climate simulations were to be performed using a coupled atmosphere-ocean configuration of CESM, these differences in the radiative effects produced by MAM3 and MARC would likely lead to differences in the climate response. In particular, the differences in the regional distribution of the radiative effects would likely impact rainfall patterns (Wang, 2015).

In light of these results, we conclude that the specific representation of aerosols in global climate models has important implications for climate modelling. Important interrelated factors include the representation of aerosol mixing state, size distribution, and optical properties.

## Appendix A: Computational performance

In order to assess the computational performance of MARC, in comparison with MAM, we have performed six timing simulations. The configuration of these simulations is described in the caption of Table S5.

Before looking at the results, it is worth noting that the default radiation diagnostics differ between MARC and MAM. As highlighted by Ghan (Ghan, 2013), in order to calculate the direct radiative effect of aerosols, a second radiation

call is required in order to diagnose "clean-sky" fluxes – in this diagnostic clean-sky radiation call, interactions between aerosols and radiation are switched off. In MARC, these clean-sky fluxes are diagnosed by default. However, in MAM, these clean-sky fluxes are not diagnosed by default, although simulations can be configured to include the necessary diagnostics. The inclusion of the clean-sky diagnostics increases computational expense. Hence, in order to facilitate a fair
comparison between MARC and MAM, we have performed two simulations for each aerosol module: one with clean-sky diagnostics switched on, and one with clean-sky diagnostics switched off.

The results from the timing simulations are shown in Table S5. When clean-sky diagnostics are switched off, as would ordinarily be the case for long climate-scale simulations, using MARC increases the computational cost by only 6% compared with a default configuration using MAM3. MAM7 is considerably more expensive. When clean-sky diagnostics
are switched on – as is the case for the simulations analysed in this manuscript – the computational cost of MARC is very similar to that of MAM3.

**Code and data availability**

CESM 1.2.2 is available via http://www.cesm.ucar.edu/models/cesm1.2/. The version of MARC used in this study is MARC v1.0.4, archived at https://doi.org/10.5281/zenodo.1117370 (Avramov et al., 2017). Model namelist files, configuration
scripts, and analysis code are available via https://github.com/grandey/p17c-marc-comparison/, archived at https://doi.org/10.5281/zenodo.1346707. The MARC input data and the model output data analysed in this paper are archived at https://doi.org/10.6084/m9.figshare.5687812.

**Author contributions**

AA and DR coupled MARC to CAM5.3 in CESM1.2.2, under the supervision of CW. AA, DR, QJ, and CW contributed to
further development of CAM5.3-MARC-ARG, with DR being the primary software maintainer. HHL and BSG contributed to testing of CAM5.3-MARC-ARG. SA contributed dust model code, optical tables, the soil erodibility map, and advice about model configuration. XL led development of MAM3 and MAM7. ZL and XL provided advice about model configuration, especially MAM7. BSG and DR designed the experiment, with contributions from QJ and CW. BSG configured and performed the simulations. BSG, DR, and HHL analysed the results. BSG produced the figures shown in
this manuscript. BSG wrote the manuscript, with contributions from all other co-authors. CW provided supervisory guidance throughout the project.

**Acknowledgements**

This research is supported by the National Research Foundation of Singapore under its Campus for Research Excellence and Technological Enterprise programme. The Center for Environmental Sensing and Modeling is an interdisciplinary research group of the Singapore-MIT Alliance for Research and Technology. This research is also supported by the U.S. National
5   Science Foundation (AGS-1339264) and the U.S. Department of Energy, Office of Science (DE-FG02-94ER61937). The CESM project is supported by the National Science Foundation and the Office of Science (BER) of the U.S. Department of Energy. We acknowledge high-performance computing support from Cheyenne (doi:10.5065/D6RX99HX) provided by NCAR's Computational and Information Systems Laboratory, sponsored by the National Science Foundation. We thank Natalie Mahowald for contributing dust model code, optical tables, a soil erodibility map, and advice, all of which have
10   aided the development of CAM5.3-MARC-ARG.

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

**Tables**

**Table 1: Global mass budgets and lifetimes of sulfate aerosol for the year-2000 MARC simulation, showing values for total sulfate aerosol and sulfate aerosol in each mode. Negative values indicate sinks. Standard errors are calculated using the annual global total for each simulation year; for the each lifetime (calculated as *Burden* divided by *Sinks*), the combined standard error is calculated following** Hogan (2006)**. The processes acting as sources and sinks are summarised in Fig. 1a. Growth and coagulation transfer sulfate between modes, but do not directly influence the total sulfate aerosol mass. Due to hydrometeor evaporation, scavenging does not necessarily result in permanent removal from the atmosphere in the form of wet deposition. (It is not possible to calculate the net wet deposition rate, because aqueous oxidation of sulfur dioxide and scavenging of gas-phase sulfuric acid also contribute to the sulfate dissolved in the hydrometeors, as shown in Fig. 1a.) Corresponding results for the year-1850 MARC simulation are shown in Table S1.**

| | Sulfate aerosol | in NUC | in AIT | in ACC | in MOS | in MBS |
|---|---|---|---|---|---|---|
| Sources, Tg(SO4)/yr | +538.70 ± 0.75 | +0.01 ± 0.00 | +0.09 ± 0.00 | +509.28 ± 0.75 | +26.45 ± 0.04 | +6.46 ± 0.03 |
| Binary nucleation | +0.00 ± 0.00 | +0.00 ± 0.00 | | | | |
| Condensation | +11.55 ± 0.05 | +0.01 ± 0.00 | +0.08 ± 0.00 | +4.37 ± 0.01 | +3.84 ± 0.02 | +3.25 ± 0.02 |
| Aging (source) | +22.33 ± 0.02 | | | | +20.59 ± 0.02 | +1.75 ± 0.00 |
| Growth (source) | | | +0.01 ± 0.00 | +0.09 ± 0.00 | | |
| Coagulation (source) | | | +0.00 ± 0.00 | +0.00 ± 0.00 | +2.02 ± 0.01 | +1.47 ± 0.01 |
| Hydrometeor evaporation | +504.82 ± 0.75 | | | +504.82 ± 0.75 | | |
| Sinks, Tg(SO4)/yr | -538.88 ± 0.74 | -0.01 ± 0.00 | -0.09 ± 0.00 | -509.37 ± 0.75 | -26.52 ± 0.04 | -6.48 ± 0.03 |
| Growth (sink) | | -0.01 ± 0.00 | -0.09 ± 0.00 | | | |
| Coagulation (sink) | | -0.00 ± 0.00 | -0.00 ± 0.00 | -3.49 ± 0.02 | | |
| Nucleation scavenging by stratiform clouds | -396.60 ± 0.69 | -0.00 ± 0.00 | -0.00 ± 0.00 | -379.23 ± 0.68 | -14.19 ± 0.02 | -3.17 ± 0.01 |
| Nucleation scavenging by convective clouds | -20.75 ± 0.06 | -0.00 ± 0.00 | -0.00 ± 0.00 | -20.08 ± 0.06 | -0.56 ± 0.00 | -0.11 ± 0.00 |
| Impaction scavenging | -116.29 ± 0.06 | -0.00 ± 0.00 | -0.00 ± 0.00 | -102.21 ± 0.06 | -11.08 ± 0.02 | -3.00 ± 0.01 |
| Dry deposition | -5.25 ± 0.01 | -0.00 ± 0.00 | -0.00 ± 0.00 | -4.34 ± 0.01 | -0.70 ± 0.00 | -0.20 ± 0.00 |
| Burden, Tg(SO4) | 1.33 ± 0.00 | 0.00 ± 0.00 | 0.00 ± 0.00 | 0.94 ± 0.00 | 0.32 ± 0.00 | 0.07 ± 0.00 |
| Lifetime, days | 0.90 ± 0.00 | 0.13 ± 0.00 | 0.09 ± 0.00 | 0.67 ± 0.00 | 4.39 ± 0.02 | 3.99 ± 0.04 |

**Table 2: Global mass budgets and lifetimes of organic carbon aerosol for the year-2000 MARC simulation, showing values for total organic carbon aerosol and organic carbon aerosol in each mode. The processes acting as sources and sinks are summarised in Fig. 1b. Emissions include volatile organic compounds converted to organic carbon aerosol. Aging and coagulation transfer organic carbon between modes, but do not directly influence the total organic carbon aerosol mass. Corresponding results for the year-1850 MARC simulation are shown in Table S2.**

|  | Organic carbon aerosol | in pure OC | in MOS |
|---|---|---|---|
| Sources, Tg/yr | +108.88 ± 0.00 | +108.88 ± 0.00 | +23.81 ± 0.02 |
| Emission | +108.88 ± 0.00 | +108.88 ± 0.00 | |
| Aging (source) | | | +21.60 ± 0.02 |
| Coagulation (source) | | | +2.22 ± 0.01 |
| Sinks, Tg/yr | -108.82 ± 0.01 | -108.75 ± 0.01 | -23.88 ± 0.02 |
| Aging (sink) | | -21.60 ± 0.02 | |
| Coagulation (sink) | | -2.22 ± 0.01 | |
| Nucleation scavenging by stratiform clouds | -13.14 ± 0.01 | -0.00 ± 0.00 | -13.14 ± 0.01 |
| Nucleation scavenging by convective clouds | -0.51 ± 0.00 | -0.00 ± 0.00 | -0.51 ± 0.00 |
| Impaction scavenging | -90.65 ± 0.02 | -81.03 ± 0.02 | -9.63 ± 0.02 |
| Dry deposition | -4.51 ± 0.00 | -3.91 ± 0.00 | -0.60 ± 0.00 |
| Burden, Tg | 1.49 ± 0.00 | 1.20 ± 0.00 | 0.29 ± 0.00 |
| Lifetime, days | 5.01 ± 0.01 | 4.03 ± 0.01 | 4.46 ± 0.02 |

**Table 3: Global mass budgets and lifetimes of black carbon aerosol for the year-2000 MARC simulation, showing values for total black carbon aerosol and black carbon aerosol in each mode. The processes acting as sources and sinks are summarised in Fig. 1c. Aging transfers black carbon between modes, but does not directly influence the total black carbon aerosol mass. Corresponding results for the year-1850 MARC simulation are shown in Table S3.**

| | Black carbon aerosol | in pure BC | in MBS |
|---|---|---|---|
| Sources, Tg/yr | +7.76 ± 0.00 | +7.76 ± 0.00 | +1.83 ± 0.00 |
|    Emission | +7.76 ± 0.00 | +7.76 ± 0.00 | |
|    Aging (source) | | | +1.83 ± 0.00 |
| Sinks, Tg/yr | -7.77 ± 0.00 | -7.76 ± 0.00 | -1.84 ± 0.00 |
|    Aging (sink) | | -1.83 ± 0.00 | |
|    Nucleation scavenging by stratiform clouds | -1.02 ± 0.00 | -0.00 ± 0.00 | -1.02 ± 0.00 |
|    Nucleation scavenging by convective clouds | -0.03 ± 0.00 | -0.00 ± 0.00 | -0.03 ± 0.00 |
|    Impaction scavenging | -6.31 ± 0.00 | -5.56 ± 0.00 | -0.74 ± 0.00 |
|    Dry deposition | -0.42 ± 0.00 | -0.36 ± 0.00 | -0.05 ± 0.00 |
| Burden, Tg | 0.11 ± 0.00 | 0.09 ± 0.00 | 0.02 ± 0.00 |
| Lifetime, days | 5.19 ± 0.01 | 4.32 ± 0.01 | 3.69 ± 0.01 |

**Table 4: Global mass budgets and lifetimes of total sea salt aerosol and total dust aerosol for the year-2000 MARC simulation, summed across the four dust aerosol modes and four sea salt aerosol modes. The processes acting as sources and sinks are summarised in Figs. 1d and 1e. Corresponding results for the year-1850 MARC simulation are shown in Table S4.**

|  | Sea salt aerosol | Dust aerosol |
|---|---|---|
| Sources, Tg/yr | +5484.88 ± 11.18 | +3683.19 ± 25.59 |
| Emission | +5484.88 ± 11.18 | +3683.19 ± 25.59 |
| Sinks, Tg/yr | -5533.62 ± 11.24 | -3705.79 ± 25.88 |
| Impaction scavenging | -2324.42 ± 4.29 | -1819.27 ± 11.26 |
| Dry deposition | -3209.19 ± 7.27 | -1886.52 ± 16.05 |
| Burden, Tg | 9.60 ± 0.02 | 40.91 ± 0.30 |
| Lifetime, days | 0.63 ± 0.00 | 4.03 ± 0.04 |

**Table 5: Area-weighted global mean radiative effects. Combined standard errors are calculated using the annual global mean for each simulation year. The regional distributions of these radiative effects are shown in Figs. 8, 14, 15, 16, and 17. $ERF_{SW+LW}$ is the sum of the other radiative effect components.**

| | 2000-1850 radiative effect, W m$^{-2}$ | | |
| --- | --- | --- | --- |
| | MAM3 | MAM7 | MARC |
| Δ direct radiative effect ($\Delta DRE_{SW}$) | - 0.02 ± 0.01 | - 0.00 ± 0.01 | - 0.17 ± 0.01 |
| Δ shortwave cloud radiative effect ($\Delta CRE_{SW}$) | - 2.09 ± 0.04 | - 2.05 ± 0.04 | - 2.17 ± 0.04 |
| Δ longwave cloud radiative effect ($\Delta CRE_{LW}$) | + 0.54 ± 0.02 | + 0.53 ± 0.02 | + 0.66 ± 0.01 |
| Δ surface albedo radiative effect ($\Delta SRE_{SW}$) | + 0.00 ± 0.02 | - 0.02 ± 0.02 | - 0.10 ± 0.02 |
| Net effective radiative forcing ($ERF_{SW+LW}$) | - 1.57 ± 0.04 | - 1.53 ± 0.04 | - 1.79 ± 0.03 |

**Figures**

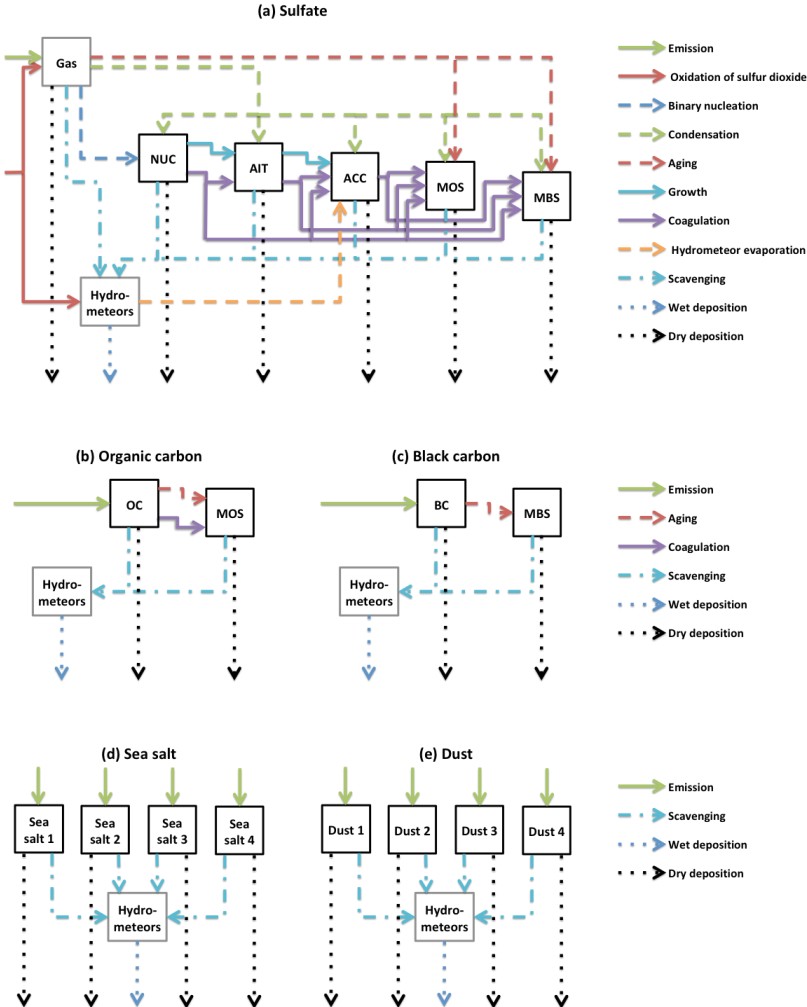

**Figure 1: Physical and chemical processes represented by MARC, organised by chemical species: (a) sulfate, (b) organic carbon, (c) black carbon, (d) sea salt, and (e) mineral dust. Black boxes represent aerosol modes: nucleation-mode sulfate (NUC), Aitken-mode sulfate (AIT), accumulation-mode sulfate (ACC), internally-mixed organic carbon plus sulfate (MOS), internally-mixed black carbon plus sulfate (MBS), pure organic carbon (OC), pure black carbon (BC), four sea salt modes, and four mineral dust modes. Grey boxes represent non-aerosol states: sulfate can exist in gas-phase (as sulfuric acid); and all five chemical species can be scavenged by hydrometeors. The arrows illustrating physical and chemical processes are listed in the legend to the right of each row. Emission of sulfate (as gas-phas sulfuric acid) and oxidation of sulfur dioxide (including both gas-phase and aqueous processes) are handled by the sulfur chemistry scheme in CAM5.3. Emission of OC includes a contribution from volatile organic compounds. For sulfate, organic carbon, and black carbon, "scavenging" includes nucleation scavenging by stratiform clouds (handled by the aerosol activation scheme) and nucleation scavenging by convective clouds (assuming a constant supersaturation of 0.1%), in addition to impaction scavenging; for sea salt and dust, "scavenging" refers to impaction scavenging only. "Dry deposition" includes gravitational settling. Further details are provided in Section 2.2 of this manuscript and Section 2.1 of Rothenberg et al. (2018).**

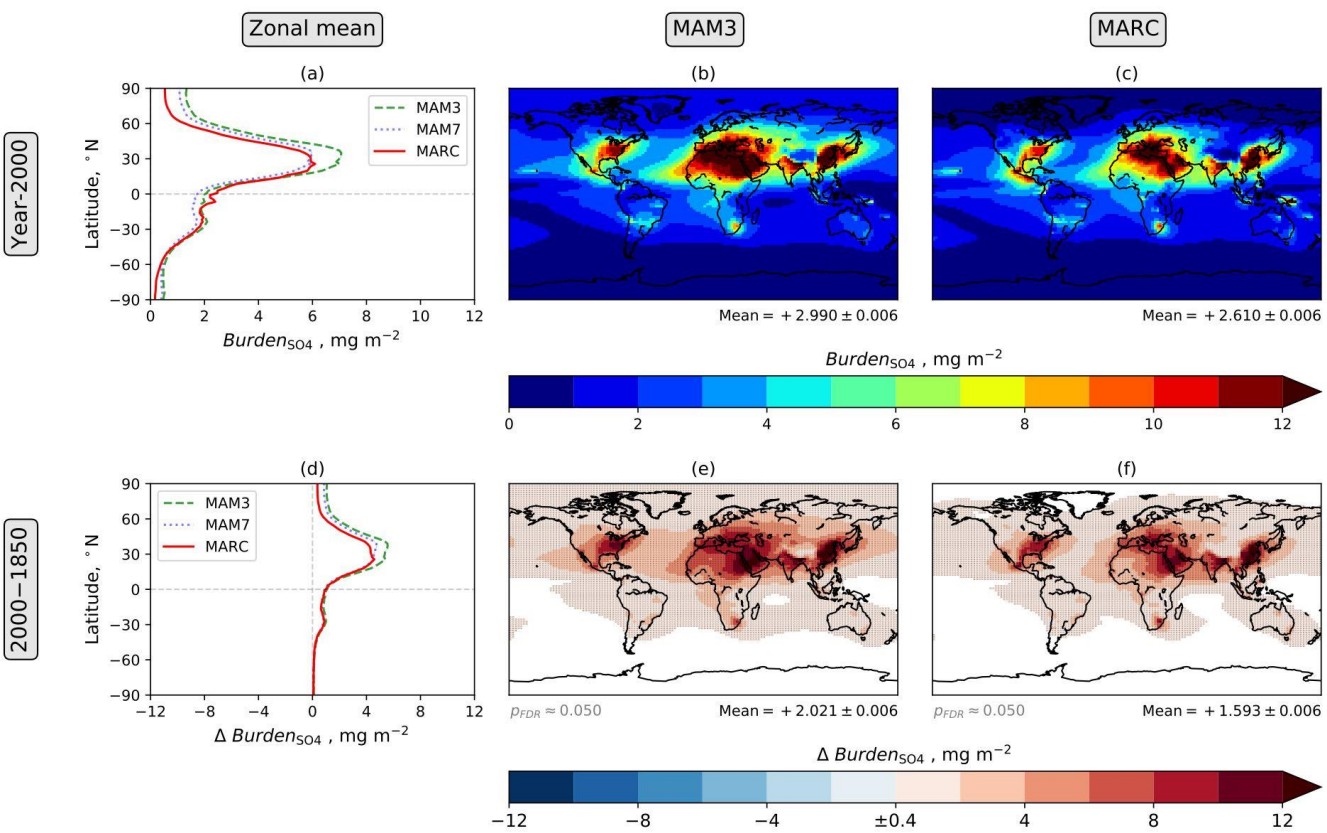

**Figure 2: Annual mean total sulfate aerosol burden ($Burden_{SO4}$).** For the zonal means (a, d), the standard errors, calculated using the annual zonal mean for each simulation year, are indicated by shading; but this shading is not visible in Fig. 2, because the standard errors are smaller than the width of the plotted lines. For the maps (b, c, e, f), the area-weighted global mean and associated standard error, calculated using the annual global mean for each simulation year, are shown below each map. For the maps showing aerosol-induced 2000-1850 differences (e, f), white indicates differences with a magnitude less than the threshold value in the centre of the colour bar ($\pm 0.4$ mg m$^{-2}$). For locations where the magnitude is greater than this threshold value, stippling indicates differences that are statistically significant at a significance level of 0.05 after controlling the false discovery rate (Benjamini and Hochberg, 1995; Wilks, 2016); the two-tailed *p* values are generated by Welch's unequal variances *t*-test, using annual mean data from each simulation year as the input; the approximate *p* value threshold, $p_{FDR}$, which takes the false discovery rate into account, is written below each map. The analysis period is 30 years. For MARC, the aerosol burdens (Figs. 2–6) have been calculated using monthly mass-mixing ratios.

**Total organic matter/carbon aerosol column burden ($Burden_{OM}$ for MAM; $Burden_{OC}$ for MARC)**

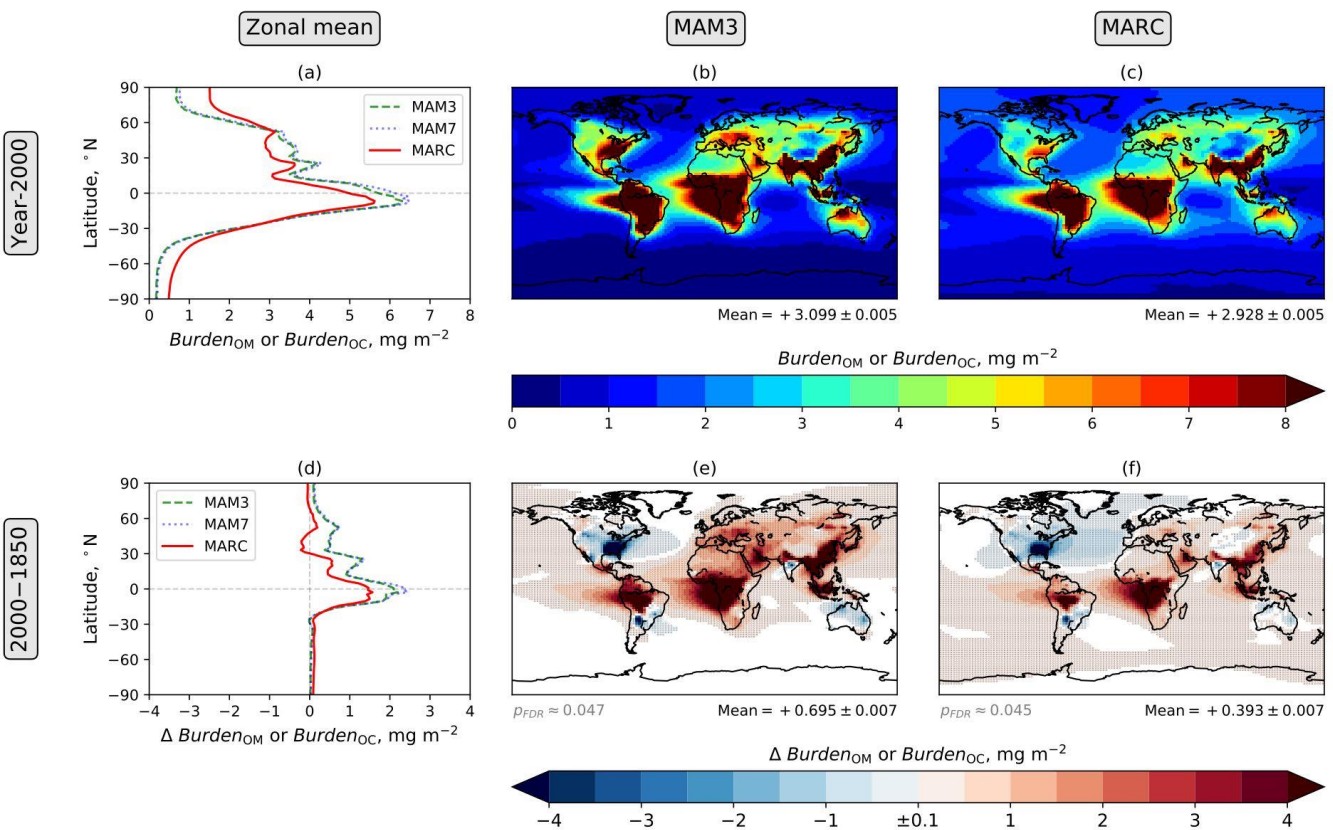

**Figure 3:** Annual mean total organic matter aerosol burden ($Burden_{OM}$) for MAM and total organic carbon aerosol burden ($Burden_{OC}$) for MARC. Due to different handling of organic carbon, MAM's $Burden_{OM}$ is not directly equivalent to MARC's $Burden_{OC}$. Secondary organic aerosol from volatile organic compounds contributes to both $Burden_{OM}$ and $Burden_{OC}$. The figure components are explained in the Fig. 2 caption.

**Total black carbon aerosol column burden ($Burden_{BC}$)**

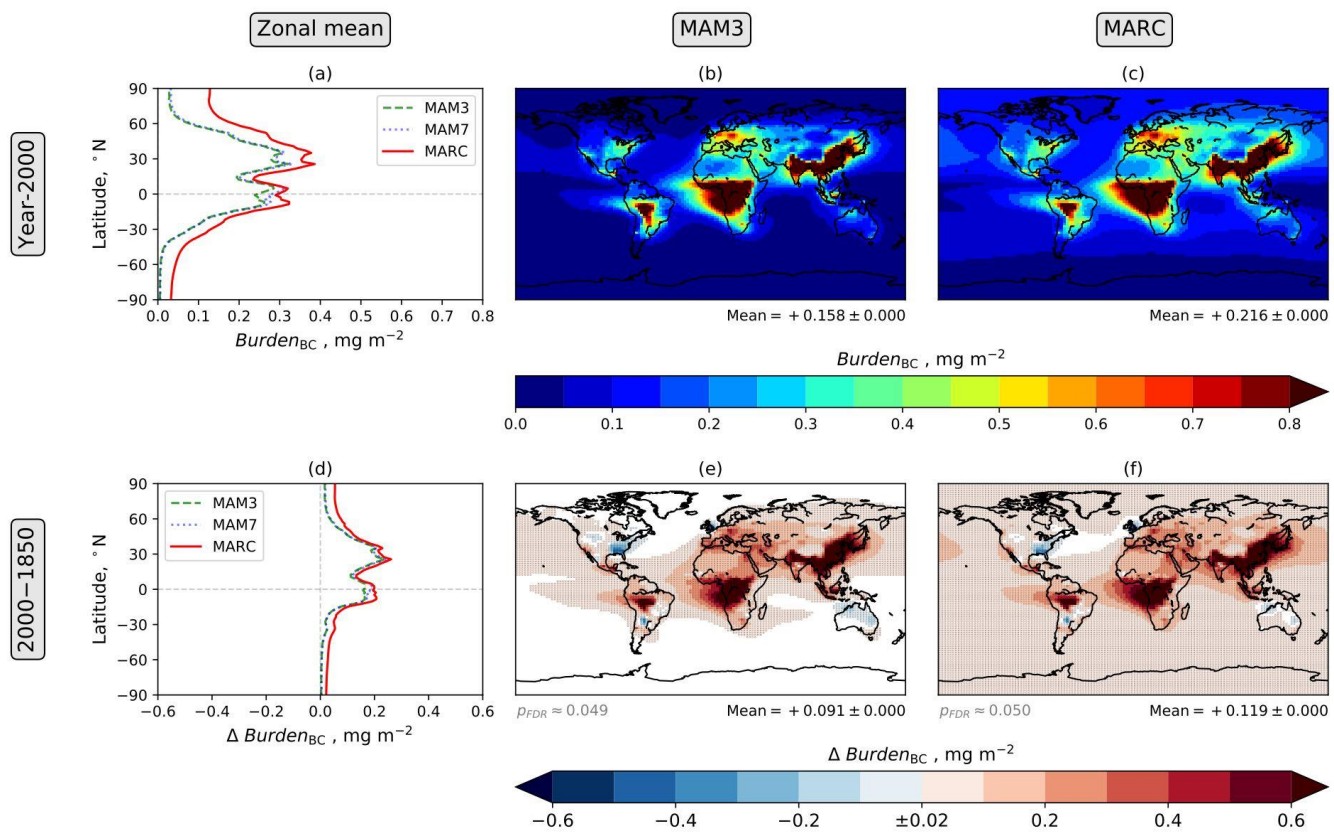

**Figure 4: Annual mean total black carbon aerosol burden ($\boldsymbol{Burden_{BC}}$). The figure components are explained in the Fig. 2 caption.**

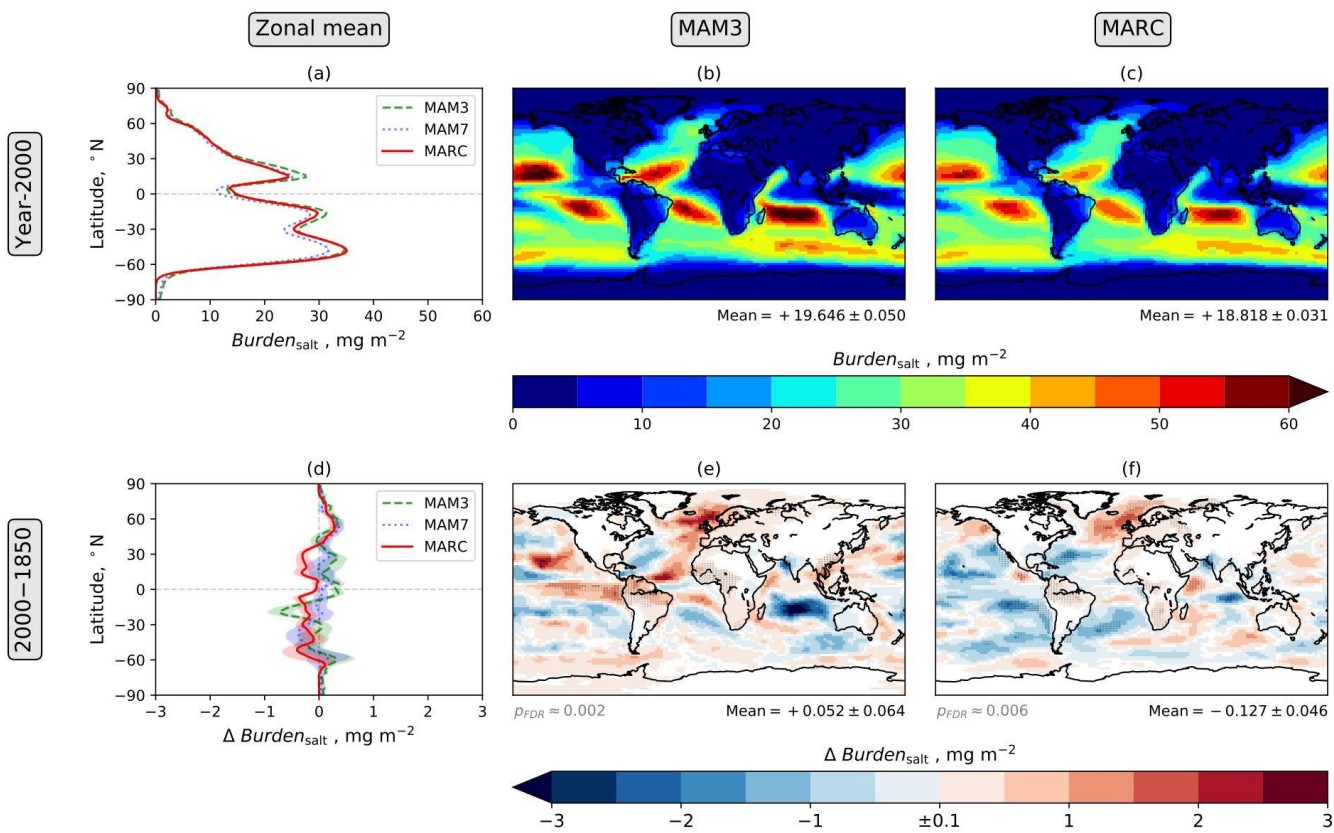

**Figure 5: Annual mean total sea salt aerosol burden ($Burden_{salt}$). The figure components are explained in the Fig. 2 caption.**

**Total dust aerosol column burden ($Burden_{dust}$)**

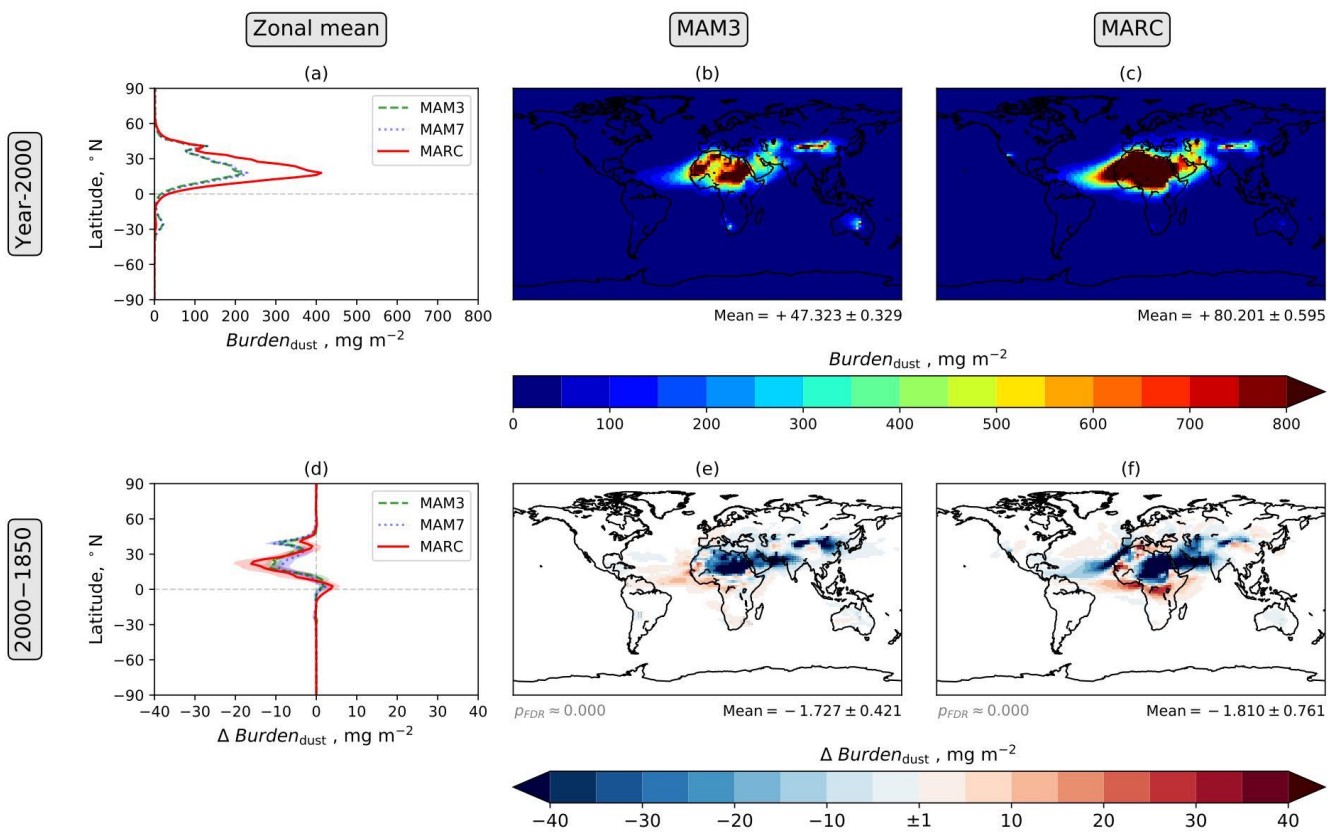

Figure 6: Annual mean dust aerosol burden ($Burden_{dust}$). The figure components are explained in the Fig. 2 caption.

# Aerosol optical depth (*AOD*)

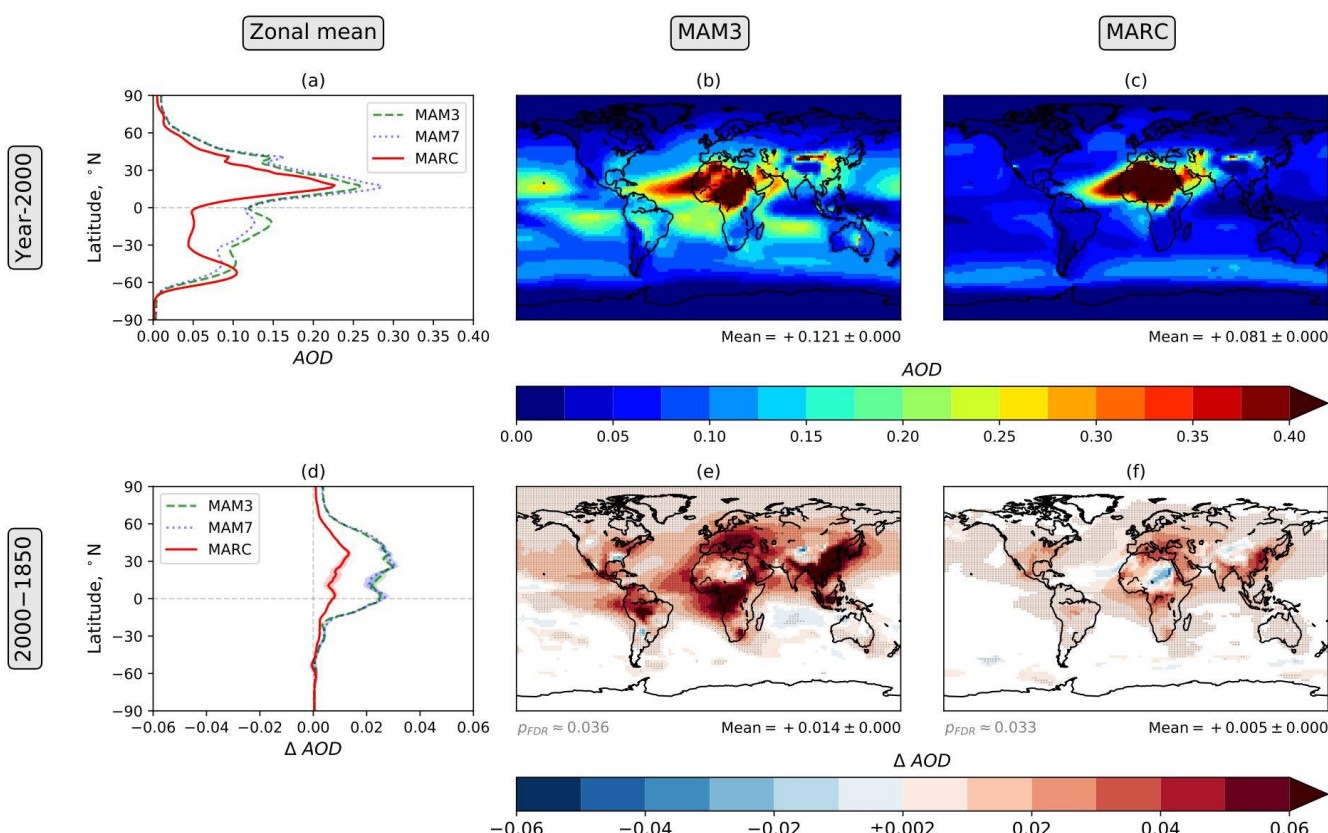

**Figure 7: Annual mean aerosol optical depth (*AOD*). The figure components are explained in the Fig. 2 caption.**

**Direct radiative effect ($DRE_{SW}$)**

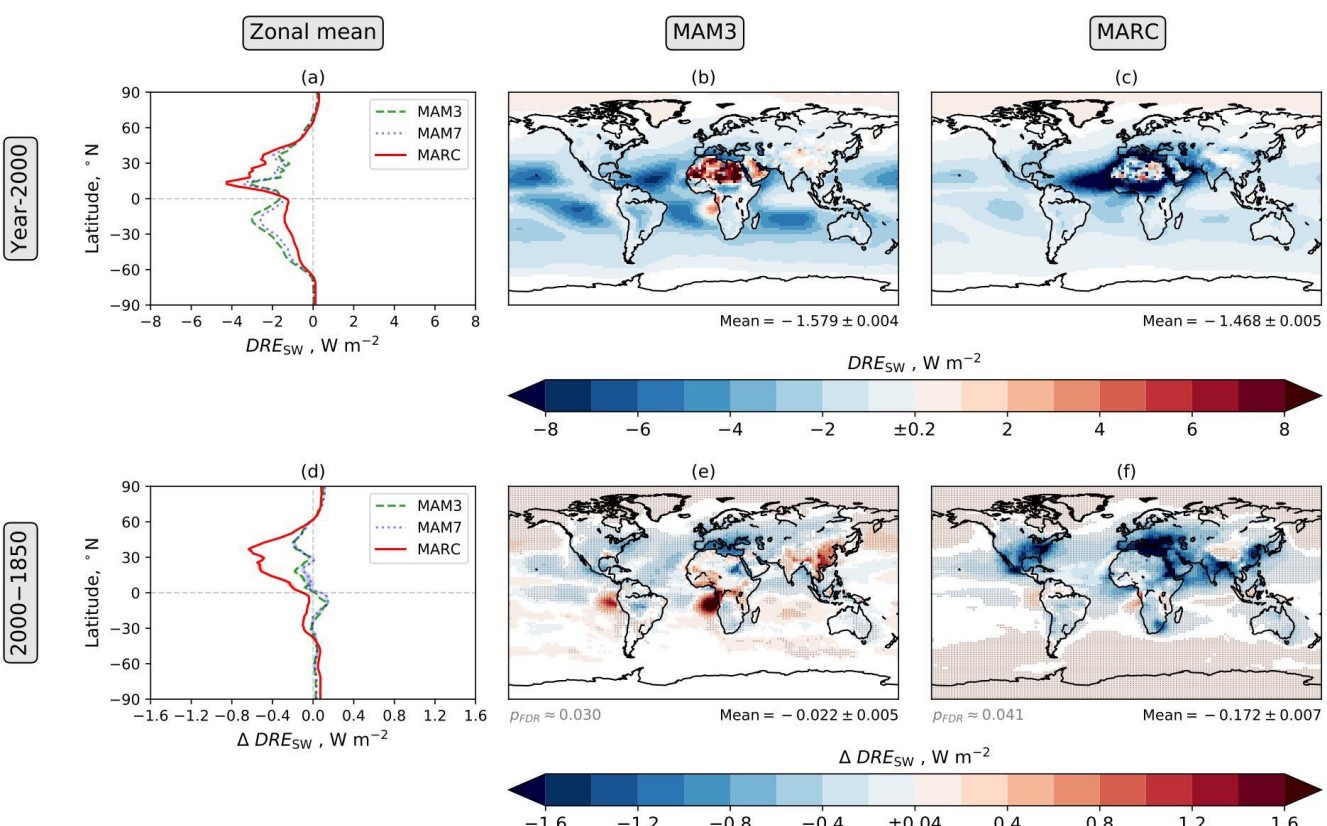

Figure 8: Annual mean direct radiative effect ($DRE_{SW}$; Eq. (3)). The figure components are explained in the Fig. 2 caption. For all four maps, white indicates differences with a magnitude less than the threshold value in the centre of the corresponding colour bar.

**Absorption by aerosols in the atmosphere ($AAA_{SW}$)**

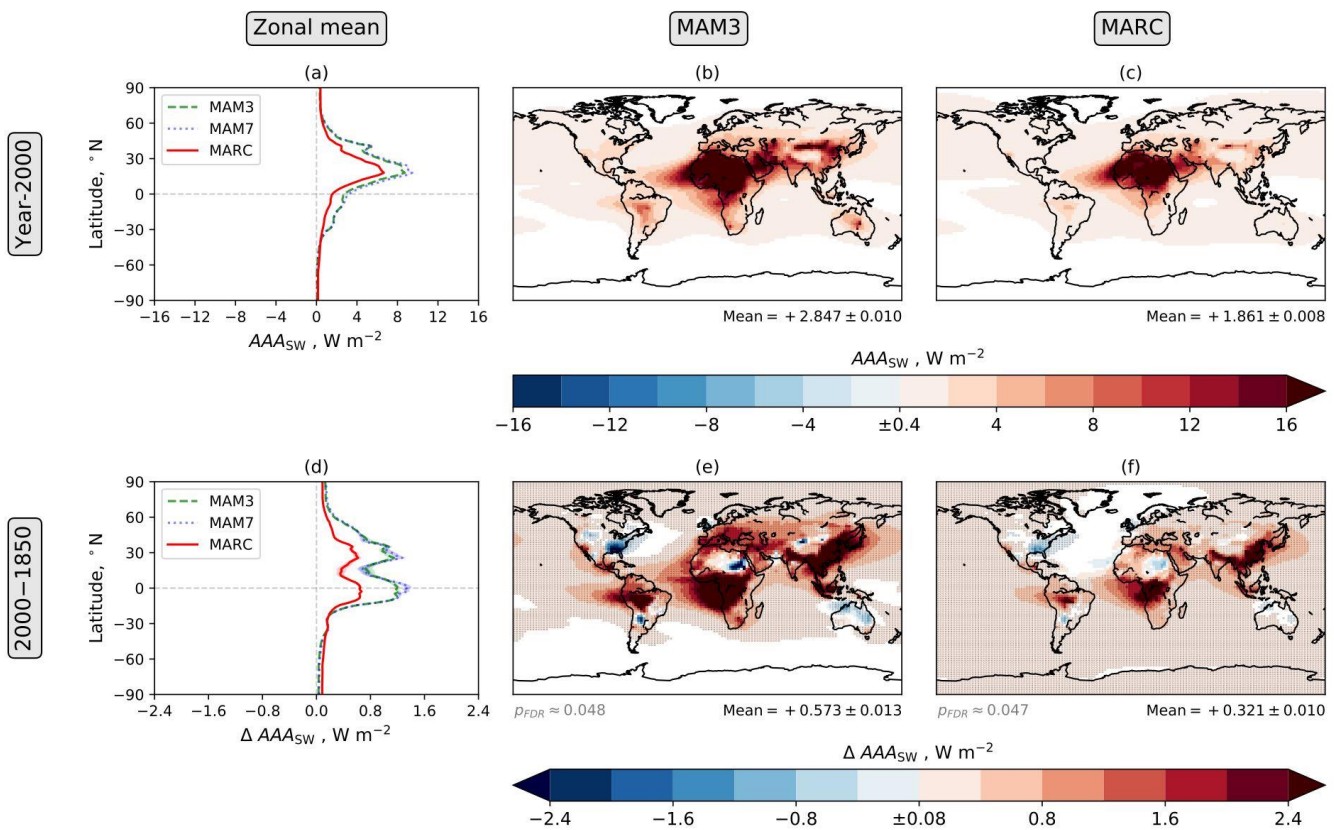

Figure 9: **Annual mean absorption by aerosols in the atmosphere ($AAA_{SW}$; Eq. (8)). The figure components are explained in the Fig. 2 caption. For all four maps, white indicates differences with a magnitude less than the threshold value in the centre of the corresponding colour bar.**

**Cloud condensation nuclei concentration at 0.1% supersaturation ($CCN_{conc}$) in model level 24 (~860hPa)**

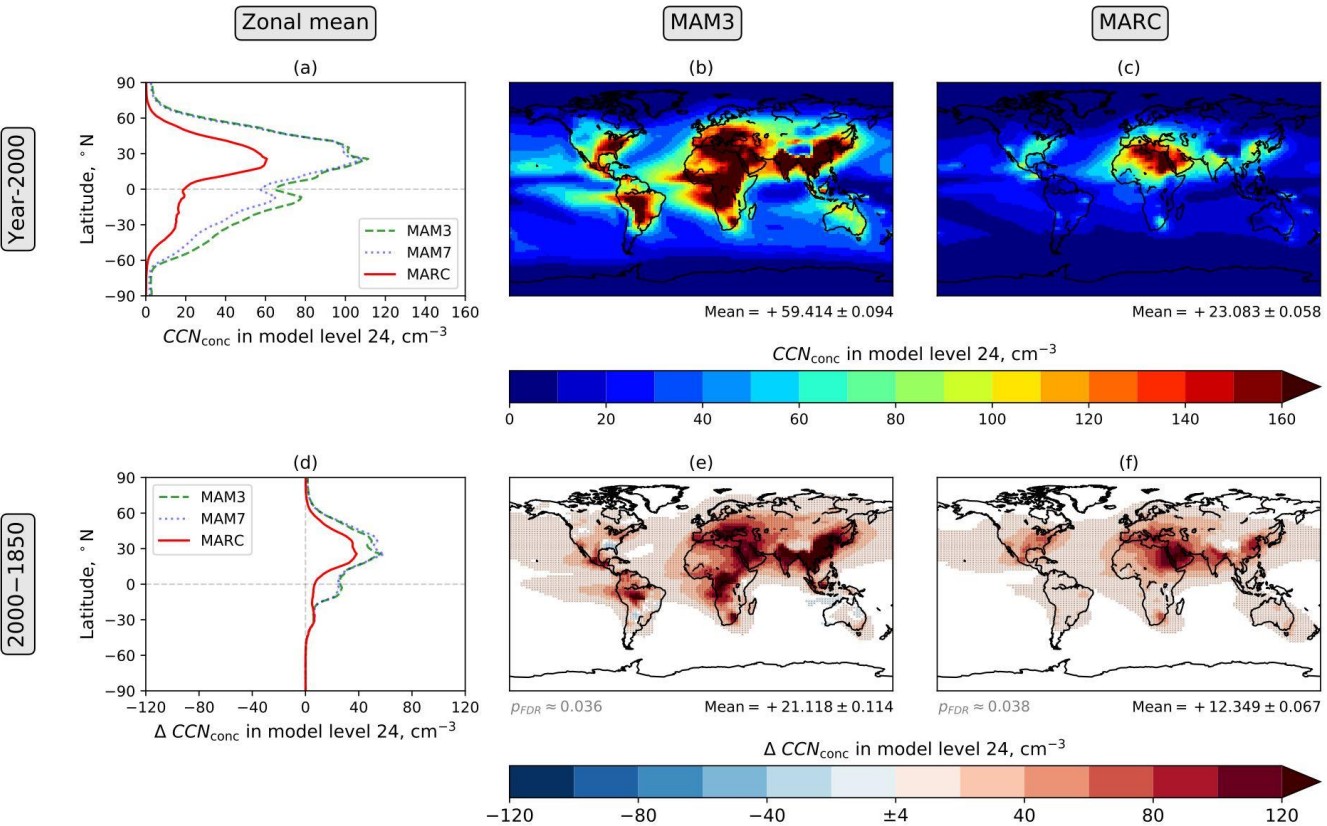

Figure 10: Annual mean cloud condensation nuclei concentration at 0.1% supersaturation ($CCN_{conc}$) in model level 24 (in the lower troposphere). The figure components are explained in the Fig. 2 caption. Corresponding results showing $CCN_{conc}$ near the surface and in the mid-troposphere are shown in Figs. S1 and S2 of the Supplement.

**Column-integrated cloud droplet number concentration ($CDNC_{column}$)**

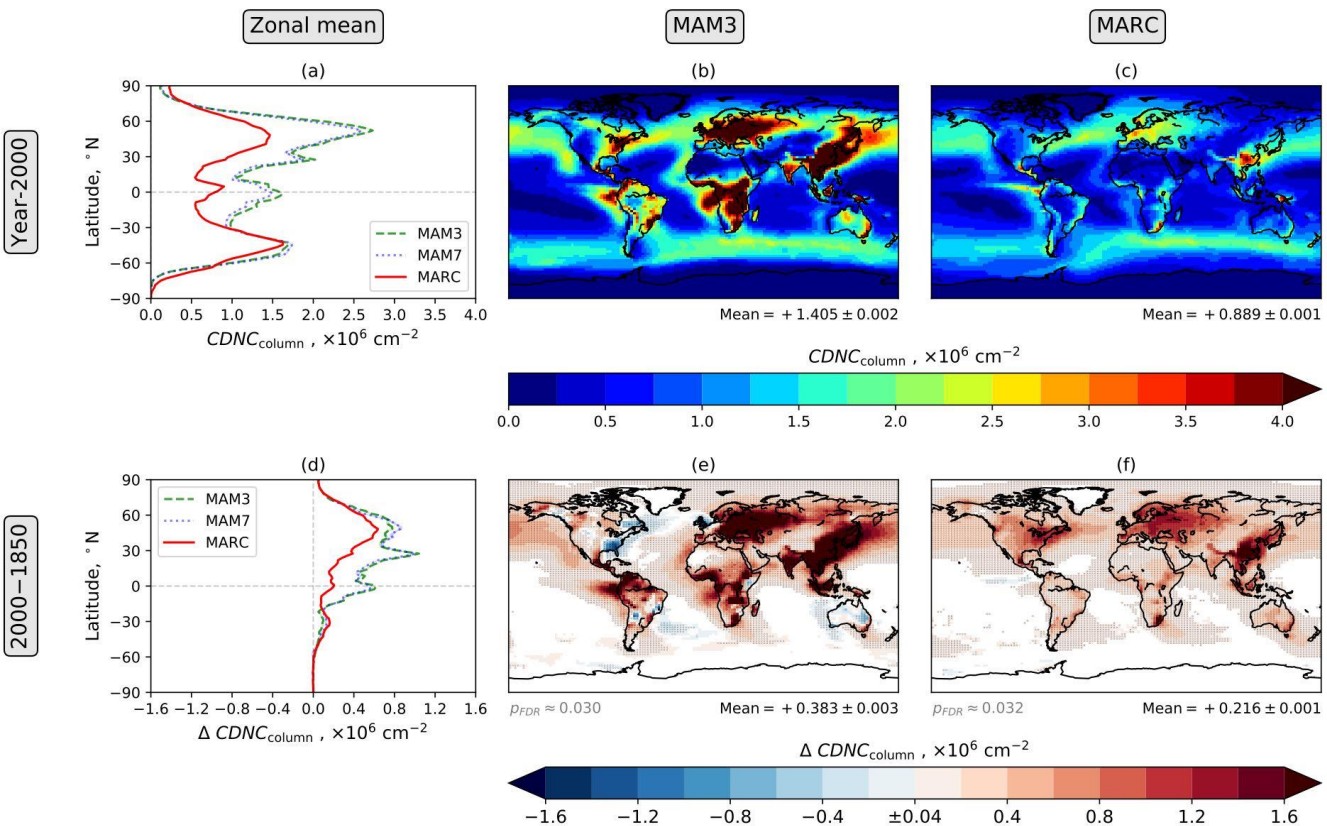

Figure 11: Annual mean column-integrated cloud droplet number concentration ($CDNC_{column}$). The figure components are explained in the Fig. 2 caption.

**Grid-box liquid water path ($WP_{liquid}$)**

**Figure 12: Annual mean grid-box cloud liquid water path ($WP_{liquid}$). The figure components are explained in the Fig. 2 caption.**

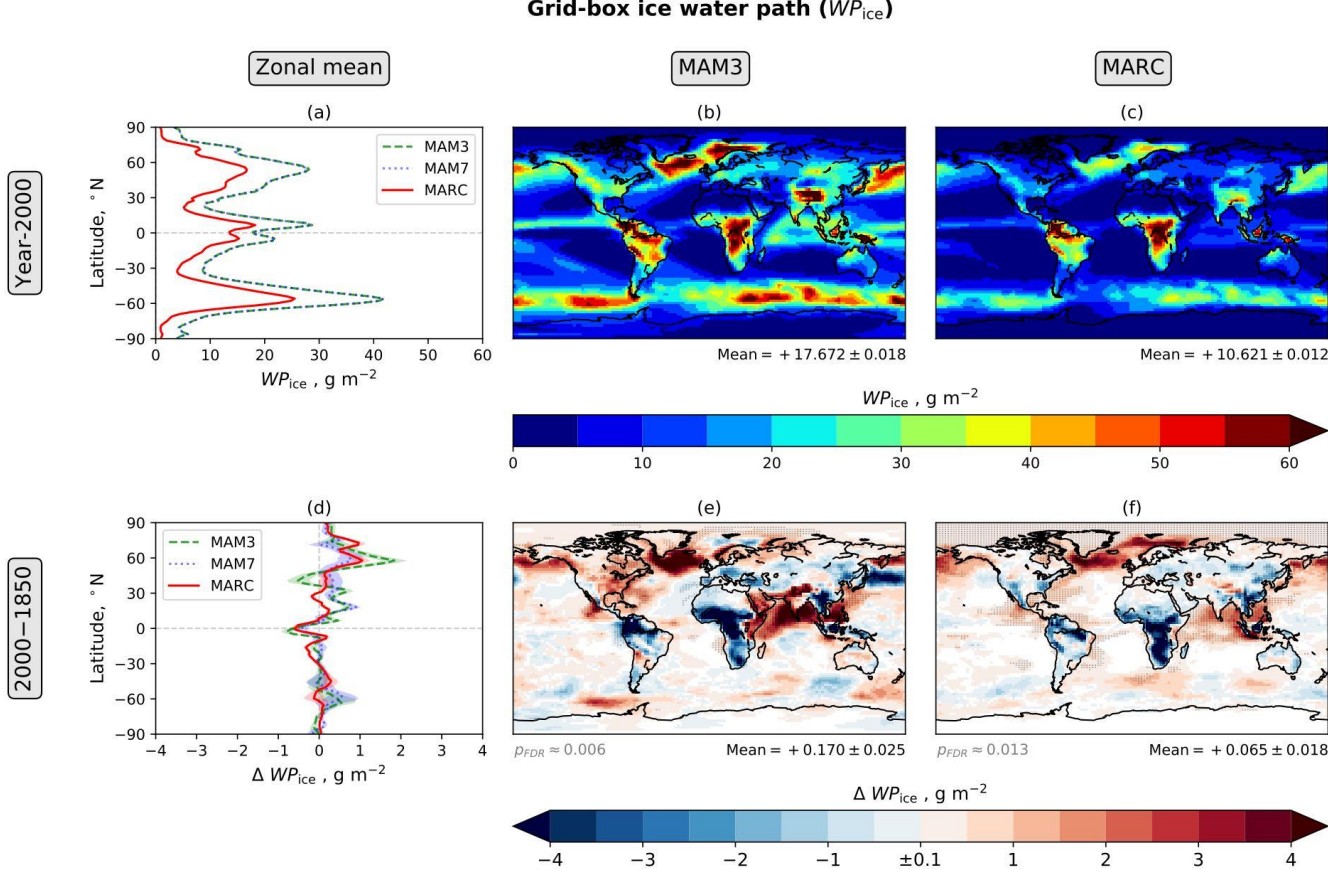

**Figure 13: Annual mean grid-box cloud ice water path ($WP_{ice}$). The figure components are explained in the Fig. 2 caption.**

**Shortwave cloud radiative effect ($CRE_{SW}$)**

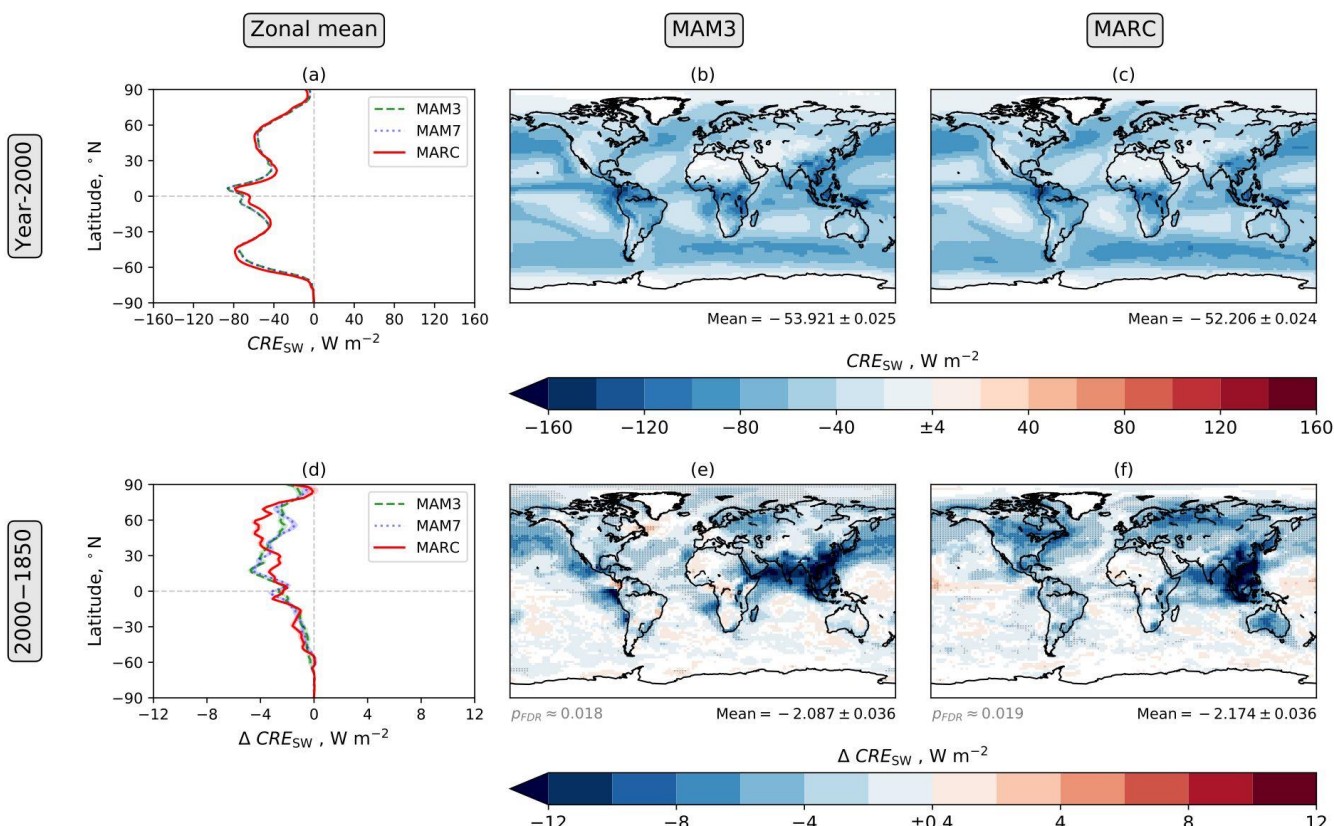

**Figure 14:** Annual mean clean-sky shortwave cloud radiative effect ($CRE_{SW}$; Eq. (4)). The figure components are explained in the Fig. 2 caption. For all four maps, white indicates differences with a magnitude less than the threshold value in the centre of the corresponding colour bar.

**Longwave cloud radiative effect ($CRE_{LW}$)**

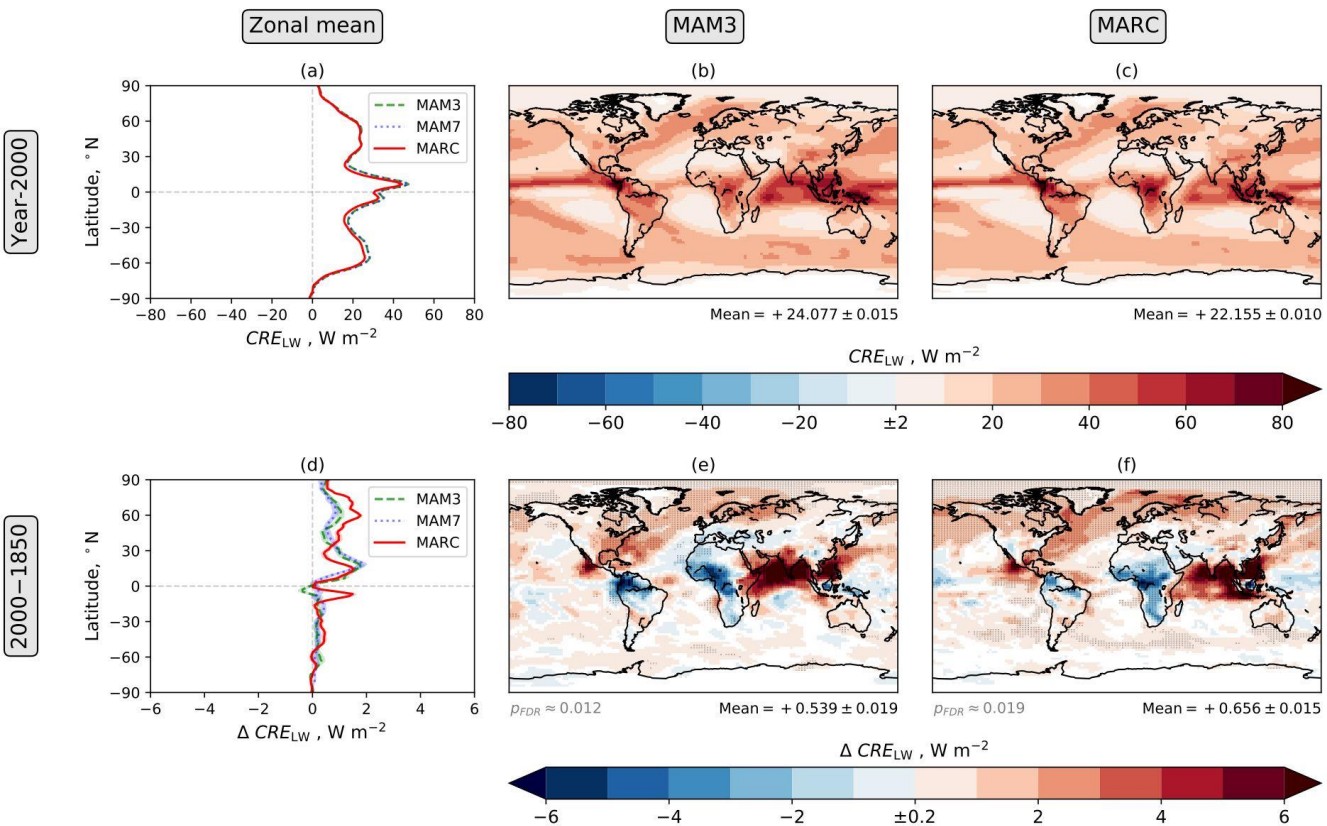

**Figure 15:** Annual mean longwave cloud radiative effect ($CRE_{LW}$; Eq. (6)). The figure components are explained in the Fig. 2 caption. For all four maps, white indicates differences with a magnitude less than the threshold value in the centre of the corresponding colour bar.

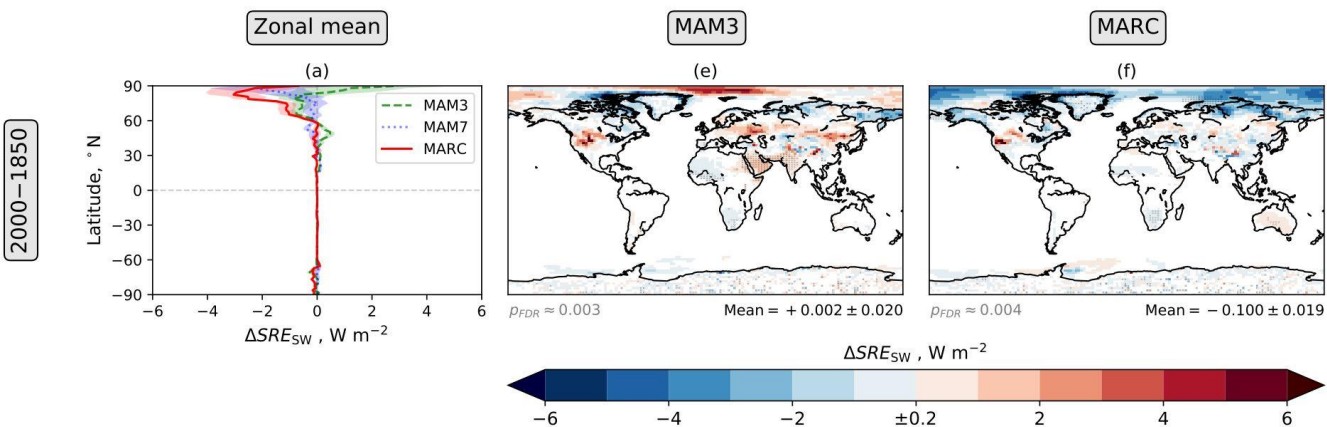

**Figure 16: Annual mean aerosol-induced 2000-1850 surface albedo radiative effect ($\Delta SRE_{SW}$; Eq. (5)). The figure components are explained in the Fig. 2 caption.**

## Net effective radiative forcing ($ERF_{SW+LW}$)

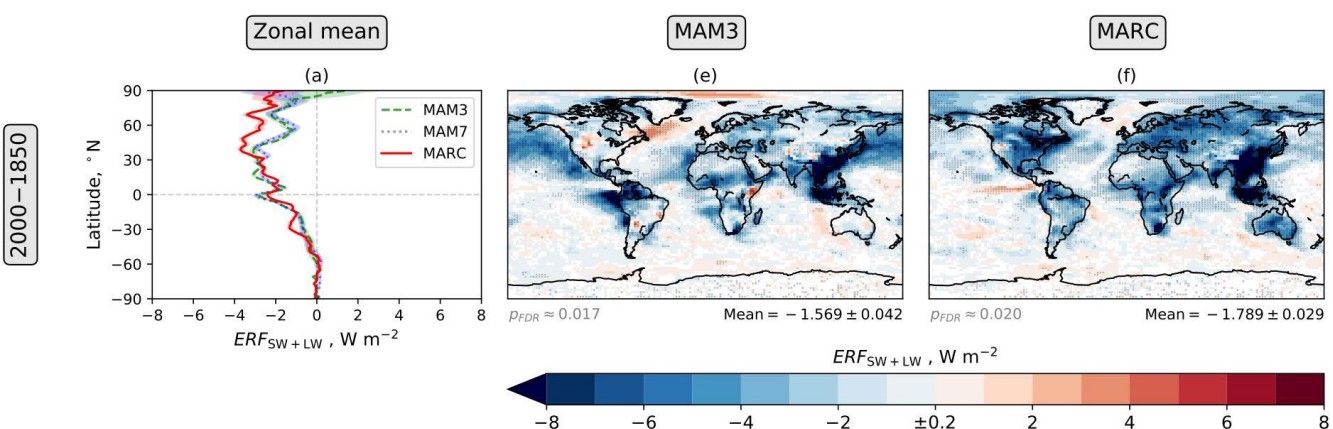

**Figure 17:** Annual mean 2000-1850 net effective radiative forcing ($ERF_{SW+LW}$; Eq. (7)). The figure components are explained in the Fig. 2 caption. When comparing the relative contributions of the different radiative effect components to $ERF_{SW+LW}$, note that different colour bars are used in Figs. 8, 14, 15, and 16.

