# Peer review of "Effective radiative forcing in the aerosol-climate model CAM5.3-MARC-ARG"

_Atmospheric Chemistry and Physics, 2018_

## Referee Comment (RC1) · Anonymous Referee #1 · 28 May 2018

Review of "Effective radiative forcing in the aerosol–climate model CAM5.3-MARC-ARG" by Benjamin S. Grandey et al.

This manuscript runs two different aerosol models in a General Circulation Model and analyzes the results with respect to aerosols and their radiative effects, specifically the anthropogenic radiative effects. The manuscript is well written and like suitable for publication in ACP with some important revisions.

There is also zero comment on which of the treatments might be more realistic, and zero justification against observations. The CCN and CDNC differences are large enough to comment on perhaps with respect to observations?

[Figure]

It is also disconcerting that of the two aerosol models analyzed, the 'more complete' MAM7 treatment does not have a pre-industrial simulation. At the very least, 2000 emissions could be used for 1850 in that simulation where they did not exist, or they could be appropriately scaled. I think this limits some of the utility of the paper.

Specific comments are contained below. The conclusions are that there are different resulting cloud effects starting with Cloud Condensation Nuclei from the different aerosol models. But I think this needs to be analyzed in a bit more depth. It seems to me as if the externally mixed or 'pure' modes for aerosols are increasing lifetime but decreasing interactions with clouds: with lower CCN and lower drop numbers, contributing to larger percentage changes due to anthropogenic sources. I think this needs to be explored a bit more.

In short, I think there needs to be a bit more analysis of the results along the lines above.

Specific Comments:

Abstract: I think more should be devoted to differences in AOD, CCN and CDNC than radiative effects. The radiative effects are the least interesting part.

Page 4, L10: so how many modes/bins in total? Are sea salt and dust only externally mixed in their sectional size bins? Please clarify. Maybe a table?

Page 5, L8: if no MAM7 emissions files, then do you have everything for MARC? What is missing for MAM7? Nitrate? Ammonia? Couldn't you find this information?

Page 6, L3: does this follow Ghan 2013? I think it does. Please state that,or where you have deviated.

Page 12, L15: but this is a key point: the different aerosol r sprsentions lead to different CCN activation spectra. Can you focus in on this a bit? This seems to be a key difference.

Page 12, L15: Marc seems to have a larger percentage change in CCN than MAM3. That probably leads to larger CDNC change (in percent) and hence the difference in LWP and SW radiative effects

Page 12, L26: I think this is a key point that needs a bit more definition. Please elaborate on what parts of MARC are leading to different activation? Can you test whether. MARC or MAM is right with observations?

Page 13, L11: are there any differences in tunable parameters between MARC and MAM simulations for CAM?

Page 14, L8: can you relate regional differences back to specific aerosol modes?

Page 14, L14: how different is the global LW (and SW) CRE between MAM and MARC?

Page 15, L11: why is BC in snow less? Lower BC mass generally? MARC seems to have more BC?

Page 16, L12: yet with longer lifetime there is less CCN? Is this because unmixed aerosol are not CCN?

Does MARC match observations better?

Page 17, L13: can you take this farther? Which specific formulation compares better to observations?

---

## Referee Comment (RC2) · Anonymous Referee #2 · 30 May 2018

In this study, the authors compare components of the aerosol effective radiative forcing and other relevant variables simulated by the CAM model using two (or three for some variables) different aerosol schemes, MAM3 (and MAM7 for some variables) and MARC. Differences depend on the component and whether one takes a regional or global view.

The paper is well written and richly illustrated. I like the decomposition by forcing components following Ghan (2013). I also like the way relevant aerosol physical distributions are compared before moving to comparing each forcing component. But the paper has three major flaws. First, the description of effective radiative forcing needs

to be sharper and refer to recent results on adjustments. Second, the paper is very diagnostic and lacks the deep explanation of causes of differences that the reader would like to see. Finally, the authors interpret differences between the two aerosol models by guessing they may be due to differences in lifetime / hygroscopicity etc. when they should instead rely on quantifying those differences in lifetime etc. The authors have chosen to make no comparisons to observations, which can only be acceptable if they provide deep understanding of the differences between the different aerosol schemes.

To address those flaws, I recommend major revisions.

**1  Main comments**

- The paragraph starting on Page 2 Line 7 should be sharper and reflect the precise definition of effective radiative forcing. That means introducing the concept of instantaneous radiative forcing and rapid adjustments. For more information, see Sherwood et al. doi:10.1175/BAMS-D-13-00167.1 . The results of the PDRMIP project would also be useful to frame the analysis of the paper, see for example Stjern et al. doi:10.1002/2017JD027326 - that is particularly true in section 3.2.2, where rapid adjustments (semi-direct effects) are not discussed at all for the moment.

- The paper is a list of differences and similarities between MAM3 and MARC, but the causes for those differences and similarities are never identified in a convincing way. Why are sulphate burdens different in the subtropics and mid-latitudes? What justifies the low hygroscopicity in externally-mixed OC in MARC (Page 7 line 30)? Why the area of positive DREsw by biomass-burning aerosols over the Sc deck of southeastern Atlantic disappears in MARC? (Figure 7) What are the changes in single-scattering albedo or absorption aerosol optical depth that would help explain differences seen in sections 3.2.2 and 3.2.3? In section 3.3.4

(Page 14 lines 6-8), the authors need to go beyond being surprised at the agreement between globally-averaged DeltaCREsw. The pre-industrial baseline is important for DeltaCREsw. So the explanation of the "surprising" agreement must lie in differences in both DeltaCDNC and DeltaLWP and year-1850 CDNC and LWP.

In addition, it is important to remember when discussing differences in sea salt (3.1.4), mineral dust (3.1.5), and surface albedo (3.4) that 2000-1850 differences are caused by feedbacks from changes in aerosol emissions, not changes in climate. So we are really looking at the impact of internal variability on wind speeds and precipitation. In that context, statistical significance is important. It would be useful to run the model several times with different initial conditions to ascertain which of the changes are noise and which are signal. That would also answer the authors' question about the causes of mineral dust changes (Page 9 line 15).

- Section 3.1 (and all aerosol modelling papers really!) should open with a Table showing the mass budget of each aerosol component (sources: emissions and chemical production; burdens; sinks: dry and wet deposition; lifetimes). This is important because it provides clear explanations for the differences that are discussed later in the paper. Too often the authors resort to guessing what the differences are (for example: Page 7 line 27; Page 8 line 14; Page 9 lines 9-10; Page 11 line 5; Page 16 line 8) when they should be able to put hard numbers on them. In addition, publishing mass budgets allows the reader to check that the schemes are balanced!

**2 Other comments**

- Page 1, lines 15-16: The first "default" is not needed.

- Page 1, line 20: The abstract should also mention the comparison to MAM7.

- Page 2, line 4: Ghan (2013) is not the best reference for aerosol-surface interactions. More relevant would be Jiao et al doi:10.5194/acp-14-2399-2014

- Page 2, line 15: "of particular importance". The pre-industrial aerosol state has also been cited as an important driver of ERF uncertainty, see Carslaw et al. doi:10.1038/nature12674

- Page 3, line 12: "Within each of these modes" That is not strictly correct, since not all species can be present in all modes. See Figure 1 of Liu et al. 2012. I suggest rephrasing to "Depending on the mode, ..."

- Page 4, line 3: The description of MARC in the Rothenberg paper is incomplete. Is there a paper that describes the current version of the scheme in more details, including a diagram like Plate 2 of Wilson et al. doi:10.1029/2000JD000198? If not, the present paper might be a good opportunity to do it. What are the mixing assumptions for emissions? I count 7 modes for MARC, excluding mineral dust and seasalt bins. Is that correct?

- Page 4, line 18: What is the mixing state of resuspended aerosols?

- Page 5, line 8: "due to a lack of year-1850 emissions files for MAM7" That is a poor excuse. Couldn't the authors make those files?

- Page 5, line 18: "from some sources". Be more specific.

- Page 5, line 31: Could also cite Lohmann and Feichter doi:10.5194/acp-5-715-2005 where ERF calculations methods originated. See their section 7.2.

- Page 6, line 6: Should reference Ghan (2013) again here for the formulas.

- Page 7, line 4: "reveals": Aerosol column burden does not "reveal" total mass, it *is* by definition the column-integrated aerosol mass.

- Page 9, line 30: And MODIS collection 6 is also different from MODIS collection 5.1, making comparisons to satellite retrievals a moving target...

- Page 9, line 31: "insufficient to explain the differences". How can the authors tell? It could be useful to compute the ratio aerosol optical depth to total burden to see if mean aerosol optical properties have changed.

- Page 10, line 24 and Figure 7: It is surprising to see DREsw peak over the Mediterranean, but it is possible that adjustments compensate instantaneous radiative forcing over the European continent.

- Page 11, line 30 and Figure 9: The peak in CCNconc over the Middle East in MARC is also surprising. Why is it there?

- Section 3.4, Page 14 line 21: Why give up on the logic of looking at relevant distributions before looking at the forcing? It would make sense to discuss Figures S11-12-13 within that section before discussing DeltaSREsw.

- Page 14, line 27: Wouldn't aerosol-induced changes in precipitation be captured in DeltaCREsw?

---

## Author Comment (AC1) · 29 Aug 2018

**Response to interactive comments on *"Effective radiative forcing in the aerosol–climate model CAM5.3-MARC-ARG"**

*B. S. Grandey et al., June–August 2018*

We would like to thank the two anonymous referees for their helpful comments. We respond to the comments from the two referees below. Comments from the referees are quoted in *green italics*. Text from the revised manuscript is quoted in *blue italics*. In addition to responding to the comments from the referees, we also summarise some major changes to the revised manuscript.

**Summary of major changes**

When preparing the revised manuscript, we have made the following major changes:

1. A year-1850 MAM7 simulation has now been performed. 2000–1850 MAM7 zonal means are now included in the figures, and the global mean radiative effect components for MAM7 have been added to revised Table 5.
2. The MARC simulations have been re-run following a correction to the organic carbon and black carbon aerosol emissions, for consistency with the MAM simulations. (Previously, these emissions were lower than they should have been.)
3. Global mass budgets and lifetimes have been quantified for MARC, resulting in eight new tables (Tables 1–4, S1–S4) and a new results section (Section 3.1).
4. For MAM, the secondary organic aerosol burden has been included in the organic matter burden. It has also been clarified that MAM's "organic matter" aerosol burden is not a direct equivalent of MARC's "organic carbon" aerosol burden.
5. For MARC, all aerosol burdens are now calculated using the diagnosed mass-mixing ratios.

Other changes have also been made in response to the referee comments, as outlined below.

**Response to Anonymous Referee #1**

*This manuscript runs two different aerosol models in a General Circulation Model and analyzes the results with respect to aerosols and their radiative effects, specifically the anthropogenic radiative effects. The manuscript is well written and like suitable for publication in ACP with some important revisions.*

Thank you for your overall assessment.

*There is also zero comment on which of the treatments might be more realistic, and zero justification against observations. The CCN and CDNC differences are large enough to comment on perhaps with respect to observations?*

Thank you for the suggestion.  However, available observations are still insufficient to constrain the $CCN_{conc}$ and $CDNC_{column}$ fields.  In the case of $CCN_{conc}$, satellite estimates are highly unreliable, and in situ measurements are limited in terms of spatial and temporal sampling.  In the case of $CDNC_{column}$, satellite estimates are not available – although cloud-top CNDC can be estimated using the adiabatic assumption, this does not reveal $CDNC_{column}$.  Nevertheless, your comment is well received.  Hence we have added the following sentences to the CDNC results section:

*"No global observations of $CDNC_{column}$ exist.  However, satellite-based estimates of cloud-top cloud droplet number concentration ($CDNC_{top}$) can be derived using the adiabatic assumption, although the uncertainties are large.  MARC tends to underestimate $CDNC_{top}$ compared to MODIS-derived estimates (Rothenberg et al., 2018)."*

*It is also disconcerting that of the two aerosol models analyzed, the 'more complete' MAM7 treatment does not have a pre-industrial simulation. At the very least, 2000 emissions could be used for 1850 in that simulation where they did not exist, or they could be appropriately scaled. I think this limits some of the utility of the paper.*

We have addressed this limitation by performing a year-1850 MAM7 simulation. The new simulation allows calculation of 2000–1850 differences for MAM7.  The zonal mean results are included in the revised manuscript.

*Specific comments are contained below. The conclusions are that there are different resulting cloud effects starting with Cloud Condensation Nuclei from the different aerosol models. But I think this needs to be analyzed in a bit more depth. It seems to me as if the externally mixed or 'pure' modes for aerosols are increasing lifetime but decreasing interactions with clouds: with lower CCN and lower drop numbers, contributing to larger percentage changes due to anthropogenic sources. I think this needs to be explored a bit more.*

*In short, I think there needs to be a bit more analysis of the results along the lines above.*

Thank you for your helpful suggestions. We have carefully considered them when revising the manuscript, including further analysis of aerosol lifetimes.

***Specific Comments:***

*Abstract: I think more should be devoted to differences in AOD, CCN and CDNC than radiative effects. The radiative effects are the least interesting part.*

Thank you for the suggestion. However, we prefer to maintain the focus on the radiative effects. Hence, in order to keep the abstract brief and focused, we have decided not to include other results – although we mention that *"We compare the aerosol lifetimes, aerosol column burdens, cloud properties, and radiative effects…"*

*Page 4, L10: so how many modes/bins in total? Are sea salt and dust only externally mixed in their sectional size bins? Please clarify. Maybe a table?*

We have included further description of MARC in the revised manuscript, including a diagram summarizing the physical and chemical processes represented by MARC (new Fig. 1).

*Page 5, L8: if no MAM7 emissions files, then do you have everything for MARC? What is missing for MAM7? Nitrate? Ammonia? Couldn't you find this information?*

You correctly identify ammonia emissions as the reason why we did not previously perform a year-1850 MAM7 simulation. We have addressed this limitation by preparing appropriate emissions in order to perform a year-1850 simulation.

*Page 6, L3: does this follow Ghan 2013? I think it does. Please state that,or where you have deviated.*

Yes, you are correct. We have made this clearer in the revised manuscript.

*Page 12, L15: but this is a key point: the different aerosol r sprsentions lead to different CCN activation spectra. Can you focus in on this a bit? This seems to be a key difference.*

We have expanded this section as follows:
*"Differences in the representation of aerosol mixing state and hygroscopicity may lead to large differences in aerosol activation spectra. In an aerosol model such as MAM that includes only internally-mixed modes, the hygroscopicity of a given mode is derived by volume weighting through all the included aerosol species and is therefore not very sensitive to changes in the chemical composition of the mode. In contrast, MARC explicitly handles mixing state and thus hygroscopicity of each individual type of aerosol: for example, the hygroscopicity of the pure OC and pure BC modes is very low, the hygroscopicity of the MOS mode depends on the internal mixing state of organic carbon and sulfate (assuming homogeneous mixing), and the hygroscopicity of the MBS mode is as high as that of pure sulfate (assuming a core-shell model) (Rothenberg et al., 2018)."*

*Page 12, L15: Marc seems to have a larger percentage change in CCN than MAM3. That probably leads to larger CDNC change (in percent) and hence the difference in LWP and SW radiative effects*

That is a good point – thank you for pointing it out. We have added (or modified) the following sentences:
*"However, the percentage increase is larger for MARC than for MAM"* (in the CCN results section);
*"However, the percentage increase is similar between MARC and MAM"* (in the CDNC results section);
*"For MARC, in comparison with MAM3, the relatively strong $\Delta WP_{liquid}$ response is consistent with the relatively strong $\Delta CCN_{conc}$ percentage change"* (in the LWP results section); and
*"The stronger $\Delta CRE_{SW}$ response for MARC is consistent with the larger $\Delta CCN_{conc}$ percentage change for MARC compared with MAM"* (in the CRE SW results section).

*Page 12, L26: I think this is a key point that needs a bit more definition. Please elaborate on what parts of MARC are leading to different activation? Can you test whether. MARC or MAM is right with observations?*

We have added the following sentences to the CDNC results section:
*"No global observations of $CDNC_{column}$ exist. However, satellite-based estimates of cloud-top cloud droplet number concentration ($CDNC_{top}$) can be derived using the adiabatic assumption, although the uncertainties are large. MARC tends to underestimate $CDNC_{top}$ compared to MODIS-derived estimates (Rothenberg et al., 2018)."*; and
*"However, the percentage increase is comparable between MARC and MAM. As was the case for $\Delta CCN_{conc}$ for MAM3, the regional distribution of $\Delta CDNC_{column}$ appears to be associated with both $\Delta Burden_{SO4}$ and $\Delta Burden_{OC}$ whereas for MARC, the regional distribution of $\Delta CDNC_{column}$ is associated with $\Delta Burden_{SO4}$ but is not closely associated with $\Delta Burden_{OC}$."*

*Page 13, L11: are there any differences in tunable parameters between MARC and MAM simulations for CAM?*

Although minor modifications have been made when coupling the ice nucleation scheme to MARC, the tuneable parameters still follow those used by MAM (Gettelman et al., 2010; Liu et al., 2007). Your question has helped us to realize that the sentence in question is misleading. We have shortened and simplified the sentence: *"The uncertainties associated with ice nucleation are very large (Garimella et al., 2017)."*

*Page 14, L8: can you relate regional differences back to specific aerosol modes?*

This is a challenging question. Having considered this question, we have added an additional sentence to the $CRE_{SW}$ section:

*"In particular, MARC produces a weaker $\Delta CRE_{SW}$ response over the stratocumulus decks near South America and southern Africa, likely because organic carbon aerosol is not an efficient source of CCN in MARC."*

*Page 14, L14: how different is the global LW (and SW) CRE between MAM and MARC?*

We have added two sentences, summarizing global mean year-2000 $CRE_{SW}$ and $CRE_{LW}$ for MAM and MARC:
*"Globally, year-2000 $CRE_{SW}$ is $-53.9$ W m$^{-2}$ for MAM3 and $-52.2$ W m$^{-2}$ for MARC"*; and
*"Globally, year-2000 $CRE_{LW}$ is $+24.1$ W m$^{-2}$ for MAM3 and $+22.2$ W m$^{-2}$ for MARC."*

*Page 15, L11: why is BC in snow less? Lower BC mass generally? MARC seems to have more BC?*

This is *"likely due to differences in dry deposition of black carbon: the rate of dry deposition of black carbon aerosol is 0.42 Tg year$^{-1}$ in MARC (Table 3), much lower than the rate of 1.27 Tg year$^{-1}$ in MAM3 (Liu et al., 2012)"*.

*Page 16, L12: yet with longer lifetime there is less CCN? Is this because unmixed aerosol are not CCN? Does MARC match observations better?*

The pure OC and pure BC modes do indeed not act as effective CCN (as can be seen in new Tables 2–3, where nucleation scavenging by stratiform clouds is close to zero for the pure modes). Regarding observations, please see our response to your next comment below.

*Page 17, L13: can you take this farther? Which specific formulation compares better to observations?*

Liu et al. (2012) and Rothenberg et al. (2018) compare simulated aerosol fields with observations. However, we are of the opinion that currently available observations are insufficient for distinguishing between MAM and MARC, especially in the context of aerosol radiative effects and aerosol-cloud interactions. In situ measurements are limited in terms of spatial and temporal sampling; and global satellite retrievals of relevant fields (e.g. CCN) are either unavailable or unreliable.

Thank you again for your helpful comments.

**Response to Anonymous Referee #2**

*In this study, the authors compare components of the aerosol effective radiative forcing and other relevant variables simulated by the CAM model using two (or three for some variables) different aerosol schemes, MAM3 (and MAM7 for some variables) and MARC. Differences depend on the component and whether one takes a regional or global view.*

*The paper is well written and richly illustrated. I like the decomposition by forcing components following Ghan (2013). I also like the way relevant aerosol physical distributions are compared before moving to comparing each forcing component. But the paper has three major flaws. First, the description of effective radiative forcing needs to be sharper and refer to recent results on adjustments. Second, the paper is very diagnostic and lacks the deep explanation of causes of differences that the reader would like to see. Finally, the authors interpret differences between the two aerosol models by guessing they may be due to differences in lifetime / hygroscopicity etc. when they should instead rely on quantifying those differences in lifetime etc. The authors have chosen to make no comparisons to observations, which can only be acceptable if they provide deep understanding of the differences between the different aerosol schemes.*

*To address those flaws, I recommend major revisions.*

Thank you for your overall assessment and suggestions. When revising the manuscript, we have addressed the flaws you identify. Further details are provided in our responses to your specific comments below.

**1 Main comments**

*The paragraph starting on Page 2 Line 7 should be sharper and reflect the precise definition of effective radiative forcing. That means introducing the concept of instantaneous radiative forcing and rapid adjustments. For more information, see Sherwood et al. doi:10.1175/BAMS-D-13-00167.1 . The results of the PDRMIP project would also be useful to frame the analysis of the paper, see for example Stjern et al. doi:10.1002/2017JD027326 - that is particularly true in section 3.2.2, where rapid adjustments (semi-direct effects) are not discussed at all for the moment.*

We have added the following sentence to the Introduction: *"In contrast to instantaneous radiative forcing, ERF allows rapid adjustments – including changes to clouds – to occur (Sherwood et al., 2015)."*

In the Results section, we have added the following sentences: *"The absorption of shortwave radiation by aerosols can drive rapid adjustments of the atmosphere, such as changes to atmospheric stability and humidity, influencing clouds (Stjern et al., 2017). Such "semidirect" effects may contribute to the cloud radiative effects discussed below. However, Ghan et al. (2012) found both the shortwave and longwave semidirect radiative effects to be statistically insignificant for MAM3 and*

*MAM7; we expect the same to apply for MARC, because the default CAM5.3 cloud microphysics scheme is also used. Cloud microphysical effects dominate the cloud radiative effects."*

*The paper is a list of differences and similarities between MAM3 and MARC, but the causes for those differences and similarities are never identified in a convincing way. Why are sulphate burdens different in the subtropics and mid-latitudes? What justifies the low hygroscopicity in externally-mixed OC in MARC (Page 7 line 30)? Why the area of positive DREsw by biomass-burning aerosols over the Sc deck of southeastern Atlantic disappears in MARC? (Figure 7) What are the changes in single-scattering albedo or absorption aerosol optical depth that would help explain differences seen in sections 3.2.2 and 3.2.3? In section 3.3.4 (Page 14 lines 6-8), the authors need to go beyond being surprised at the agreement between globally-averaged DeltaCREsw. The pre-industrial baseline is important for DeltaCREsw. So the explanation of the "surprising" agreement must lie in differences in both DeltaCDNC and DeltaLWP and year-1850 CDNC and LWP.*

We have addressed these questions in the revised manuscript.

In the sulfate burden results section, we have added the following sentences: *"The differences between MARC, MAM3, and MAM7 may be due to differences in sulfate aerosol lifetime. However, it is not possible to test this conclusively: as pointed out in Section 3.1, the sulfate aerosol lifetimes diagnosed for MARC should not be directly compared to those diagnosed for MAM, because cloud-cycling contributes to the sources and sinks of sulfate aerosol in MARC."*

Regarding the hygroscopicity of the aerosol modes in MARC, in the Introduction we now refer readers to Rothenberg et al. (2018) and Petters and Kreidenweis (2007): *"Table 1 of Rothenberg et al. (2018) contains details of the size distribution, density, and hygroscopicity of each mode. It is assumed that the pure OC and pure BC modes are hydrophobic (Petters and Kreidenweis, 2007; Rothenberg et al., 2018)."*

In the DRE results section, we have added the following sentence: *"The impact of black carbon aerosol on year-2000 $DRE_{SW}$ also differs between MAM3 and MARC: for MAM3, absorption by black carbon aerosol drives positive values of $DRE_{SW}$ over the South Atlantic stratocumulus deck near southern Africa; for MARC, negative values of $DRE_{SW}$ are found over the stratocumulus deck, suggesting weaker absorption by black carbon aerosol compared with MAM3. The differing absorption is likely due to the differing representations of aerosol mixing state and associated optical properties: the majority of MARC's black carbon aerosol is found in the pure BC mode, whereas MAM's black carbon aerosol is internally mixed with other species, likely leading to stronger absorption for MAM compared with MARC."*

In the AAA results section, we have clarified that there is *"weaker absorption for MARC"*.

In the CRE SW results section, we have modified the discussion to correspond to our updated MARC results: *"When globally averaged, the global mean $\Delta CRE_{SW}$ for*

*MARC ($-2.17 \pm 0.04$ W m$^{-2}$) is stronger than that for MAM3 ($-2.09 \pm 0.04$ W m$^{-2}$) and MAM7 ($-2.05 \pm 0.04$ W m$^{-2}$). For MARC compared with MAM, the stronger $\Delta CRE_{SW}$ response for MARC is consistent with the larger $\Delta CCN_{conc}$ percentage change."*

*In addition, it is important to remember when discussing differences in sea salt (3.1.4), mineral dust (3.1.5), and surface albedo (3.4) that 2000-1850 differences are caused by feedbacks from changes in aerosol emissions, not changes in climate. So we are really looking at the impact of internal variability on wind speeds and precipitation. In that context, statistical significance is important. It would be useful to run the model several times with different initial conditions to ascertain which of the changes are noise and which are signal. That would also answer the authors' question about the causes of mineral dust changes (Page 9 line 15).*

Thank you for highlighting the potential for confusion in the interpretation of "2000-1850". In the Section 2.3, we have clarified that *"The "2000-1850" differences should be interpreted as aerosol-induced differences, arising due to changes in aerosol emissions alone: the only difference between the year-2000 simulations and the year-1850 simulations is the aerosol (including aerosol precursor) emissions."* In order to remind readers of this, we now use the term *"aerosol-induced"* in some of the sentences in the results section and captions.

The 30-year analysis period gives us an "ensemble" of 30 annual means that we use to assess significance and calculate standard errors. When discussing the sea salt burden results, we point out that *"the aerosol-induced 2000-1850 differences in Burden$_{salt}$ surface wind speed, and precipitation rate are both relatively small and often statistically insignificant across most of the world. Hence these 2000-1850 differences may be due primarily to internal variability."*

*Section 3.1 (and all aerosol modelling papers really!) should open with a Table showing the mass budget of each aerosol component (sources: emissions and chemical production; burdens; sinks: dry and wet deposition; lifetimes). This is important because it provides clear explanations for the differences that are discussed later in the paper. Too often the authors resort to guessing what the differences are (for example: Page 7 line 27; Page 8 line 14; Page 9 lines 9-10; Page 11 line 5; Page 16 line 8) when they should be able to put hard numbers on them. In addition, publishing mass budgets allows the reader to check that the schemes are balanced!*

Thank you for this very helpful suggestion. We have diagnosed the budgets for MARC, and have included these as new Tables 1–4 and S1–S4. We have also included a new section (Section 3.1) discussing these results.

**2 Other comments**

*Page 1, lines 15-16: The first "default" is not needed.*

We have removed "default" before "ARG".

*Page 1, line 20: The abstract should also mention the comparison to MAM7.*

We now mention MAM7 in the abstract.

*Page 2, line 4: Ghan (2013) is not the best reference for aerosol-surface interactions. More relevant would be Jiao et al doi:10.5194/acp-14-2399-2014*

Thank you for pointing out the relevant reference. We have followed your suggestion.

*Page 2, line 15: "of particular importance". The pre-industrial aerosol state has also been cited as an important driver of ERF uncertainty, see Carslaw et al. doi:10.1038/nature12674*

Following your suggestion, we have added the following sentences: *"Much of the uncertainty [in the anthropogenic aerosol ERF] can be attributed to uncertainty in pre-industrial aerosol emissions (Carslaw et al., 2013). Model parameterizations constitute another large source of uncertainty."*

*Page 3, line 12: "Within each of these modes" That is not strictly correct, since not all species can be present in all modes. See Figure 1 of Liu et al. 2012. I suggest rephrasing to "Depending on the mode, …"*

We have adopted your suggestion.

*Page 4, line 3: The description of MARC in the Rothenberg paper is incomplete. Is there a paper that describes the current version of the scheme in more details, including a diagram like Plate 2 of Wilson et al. doi:10.1029/2000JD000198? If not, the present paper might be a good opportunity to do it. What are the mixing assumptions for emissions? I count 7 modes for MARC, excluding mineral dust and seasalt bins. Is that correct?*

Thank you for the suggestion to include a diagram summarizing the modes and processes. Our updated manuscript includes such a diagram (new Figure 1).

Yes, MARC contains seven modes, excluding mineral dust and sea salt bins. As mentioned in Section 2.2, *"In the MOS mode, it is assumed that the organic carbon and sulfate are mixed homogeneously within each particle; in the MBS mode, it is assumed that each particle contains a black carbon core surrounded by a sulfate shell."*

*Page 4, line 18: What is the mixing state of resuspended aerosols?*

We have modified the sentence to make this clearer: *"Evaporation of cloud and rain drops results in resuspension of sulfate aerosol in the accumulation-mode."*

*Page 5, line 8: "due to a lack of year-1850 emissions files for MAM7" That is a poor excuse. Couldn't the authors make those files?*

We have addressed this limitation by performing a year-1850 MAM7 simulation with appropriate emissions. We have included analysis of this year-1850 MAM7 simulation in the revised manuscript.

*Page 5, line 18: "from some sources". Be more specific.*

We now mention the specific sectors.

*Page 5, line 31: Could also cite Lohmann and Feichter doi:10.5194/acp-5-715-2005 where ERF calculations methods originated. See their section 7.2.*

We have added the reference, as you suggest.

*Page 6, line 6: Should reference Ghan (2013) again here for the formulas.*

We have adopted your suggestion.

*Page 7, line 4: "reveals": Aerosol column burden does not "reveal" total mass, it is by definition the column-integrated aerosol mass.*

We have replaced "reveals" with "is".

*Page 9, line 30: And MODIS collection 6 is also different from MODIS collection 5.1, making comparisons to satellite retrievals a moving target...*

We have added that *"satellite-based retrievals of AOD are not error-free"*.

*Page 9, line 31: "insufficient to explain the differences". How can the authors tell? It could be useful to compute the ratio aerosol optical depth to total burden to see if mean aerosol optical properties have changed.*

This is an intriguing suggestion. However, we do not think that looking at total burden – summed across all species – would provide much further insight.

We have modified the sentence as follows: *"The differences between the aerosol burdens for MAM and MARC, discussed above, are insufficient to explain the differences in year-2000 AOD, especially the large difference in the Southern Hemisphere subtropics (where the burdens are similar between MAM and MARC). Hence it is likely that differences in the aerosol optical properties between MARC and MAM are responsible for the fact that MARC generally produces lower values of AOD. In particular, the large difference in year-2000 AOD between MAM3 and MARC over the subtropical ocean, where $Burden_{salt}$ is large, is likely due to differences in the optical properties of sea salt aerosol."*

*Page 10, line 24 and Figure 7: It is surprising to see DREsw peak over the Mediterranean, but it is possible that adjustments compensate instantaneous radiative forcing over the European continent.*
*Page 11, line 30 and Figure 9: The peak in CCNconc over the Middle East in MARC is also surprising. Why is it there?*

For MARC, the DRE and CCN$_{conc}$ results follow a similar pattern to the sulfate burden results. The year-2000 and 2000-1850 sulfate burdens are also high over the Mediterranean and Middle East.

*Section 3.4, Page 14 line 21: Why give up on the logic of looking at relevant distributions before looking at the forcing? It would make sense to discuss Figures S11-12-13 within that section before discussing DeltaSREsw.*

Thank you for the suggestion to rearrange this section in order to discuss the relevant distributions prior to discussing $\Delta SRE_{SW}$. However, having considered your suggestion, we prefer to keep the order of this section in it's current form: due to fact that we include a paragraph introducing what $\Delta SRE_{SW}$ does and does not include at the beginning over this section, we think it makes sense to then immediately discuss the $\Delta SRE_{SW}$ results before considering the reasons for the differences.

*Page 14, line 27: Wouldn't aerosol-induced changes in precipitation be captured in DeltaCREsw?*

$\Delta CRE$ quantifies the radiative impacts of aerosols via clouds. Although $\Delta CRE$ and aerosol-induced changes in precipitation may be related, neither are they synonymous nor can one be subsumed under the other.

Thank you again for your helpful comments.

[revised manuscript text omitted]

**Mass of black carbon (BC) in top layer of snow over land**

[Figure]

Figure S13: **Annual mean mass of black carbon in the the top layer over snow over land. The figure components are explained in the caption of Fig. 2.**

**References**

CESM Software Engineering Group: CESM User's Guide (CESM1.2 Release Series User's Guide), [online] Available from: http://www.cesm.ucar.edu/models/cesm1.2/cesm/doc/usersguide/ug.pdf [accessed 2017-10-31], 2015.

---

## Author Response (AR2)

**Response to referee comments on *"Effective radiative forcing in the aerosol–climate model CAM5.3-MARC-ARG"**

*B. S. Grandey et al., October 2018*

We thank the editor and referee for reading our revised manuscript, and for recommending that the manuscript be accepted subject to minor revisions. Below, we respond to the comments from the referee (quoted in *green italics*). A revised manuscript, with changes highlighted, follows.

**Response to Anonymous Referee**

*I thank the authors for their efforts in addressing reviewers' comments so comprehensively. References to rapid adjustments are now clearer. The discussion of differences between the different schemes now tries to identify the causes for those differences, although not all differences are understood. And it is now clear that differences in mineral dust and seasalt are in fact internal variability.*

Thank you again for your earlier comments, which have enabled us to improve the manuscript. Thank you for your assessment of the revised manuscript and for your further comments.

*The addition of Tables to document lifetimes and burdens is especially welcome, and will be useful for future readers. I however do not understand why the true sulfate aerosol lifetime cannot be calculated. What the authors call "hydrometeor evaporation" is in-cloud production of sulfate, from what I understand. Then scavenging is not really scavenging because some of it is re-evaporated. The difference between the sum of the three scavenging processes and hydrometeor evaporation is -29 Tg yr-1. Isn't that the actual wet deposition? That would imply a sulfate lifetime of 14 days, though, which is long. It would really be good to have an actual sulfate lifetime estimate.*

Thank you for suggesting the possibility of calculating the net wet deposition rate and hence a revised sulfate lifetime estimate. However, it is not possible to calculate the net wet deposition rate as $\sum$(*Scavenging terms*) minus *Hydrometeor evaporation*, because aqueous oxidation of sulfur dioxide and scavenging of gas-phase sulfuric acid also contribute to the sulfate dissolved in the hydrometeors (Fig. 1a). We have now clarified this in the caption of Table 1.

*I also thank the authors for adding Figure 1 as a description of the MARC scheme. Its quality could be improved by making the lines and labels thicker.*

Thank you for your suggestion. We have now made the lines and labels thicker.

[revised manuscript text omitted]